# Oracle-Efficient Regret Minimization in Factored MDPs with Unknown Structure

**Aviv Rosenberg**
Tel-Aviv University
avivros007@gmail.com

**Yishay Mansour**
Tel-Aviv University and Google Research, Tel Aviv
mansour@tau.ac.il

## Abstract

We study regret minimization in non-episodic factored Markov decision processes (FMDPs), where all existing algorithms make the strong assumption that the factored structure of the FMDP is known to the learner in advance. In this paper, we provide the first algorithm that learns the structure of the FMDP while minimizing the regret. Our algorithm is based on the optimism in face of uncertainty principle, combined with a simple statistical method for structure learning, and can be implemented efficiently given oracle-access to an FMDP planner. Moreover, we give a variant of our algorithm that remains efficient even when the oracle is limited to non-factored actions, which is the case with almost all existing approximate planners. Finally, we leverage our techniques to prove a novel lower bound for the known structure case, closing the gap to the regret bound of Chen et al. [2021].

## 1 Introduction

Reinforcement learning (RL) considers an agent interacting with an unknown stochastic environment with the aim of maximizing its expected cumulative reward. This is usually modeled by a Markov decision process (MDP) with a finite number of states. The vast majority of provably-efficient RL has focused on the tabular case, where the state space is assumed to be small. Starting with the UCRL2 algorithm [Jaksch et al., 2010], near-optimal regret bounds were proved [Azar et al., 2017, Fruit et al., 2018, Jin et al., 2018, Zanette and Brunskill, 2019, Efroni et al., 2019]. Unfortunately, many real-world RL applications involve problems with a huge state space, yielding the tabular MDP model impractical as it requires the regret to unavoidably scale polynomially with the number of states.

In many practical scenarios, prior knowledge about the environment can be leveraged in order to develop more efficient algorithms. A popular way to model additional knowledge about the structure of the environment is by *factored MDPs* (FMDPs; Boutilier et al. [1995, 1999]). The state of an FMDP is composed of $d$ components, called *factors*, and each component is determined by only $m$ other factors, called its *scope*. FMDPs arise naturally in many applications like games, robotics, image-based applications and production lines (where only neighbouring machines affect one another). The common property of all these examples is the huge state space exponential in $d$, but the very small scope size $m$ (e.g., in images each pixel is a factor and it depends only on neighboring pixels).

The key benefit of FMDPs is the combinatorial state space that allows compact representation. That is, although the number of states is exponential in $d$, the FMDP representation is only exponential in $m$ (which is much smaller) and polynomial in $d$. Early works [Kearns and Koller, 1999, Guestrin et al., 2002, Strehl, 2007, Szita and Lőrincz, 2009] show that FMDPs also reduce the sample complexity exponentially, thus avoiding polynomial dependence on the number of states. Recently, this was further extended to algorithms with near-optimal regret bounds [Osband and Van Roy, 2014, Xu and Tewari, 2020, Tian et al., 2020, Chen et al., 2021, Talebi et al., 2021]. However, all these works make the strong assumption that the underlying FMDP structure is fully known to the learner in advance.

35th Conference on Neural Information Processing Systems (NeurIPS 2021).

In this paper we provide the first regret minimization algorithm for FMDPs with unknown structure, thus solving an open problem from Osband and Van Roy [2014]. Our algorithm is built on a novel concept of *consistent scopes* and guarantees **near-optimal $\sqrt{T}$ regret** that scales polynomially with the FMDP encoding and is therefore exponentially smaller than the number of states (and the regret in tabular MDPs). Moreover, our algorithm features an innovative construction that can incorporate elimination of inconsistent scopes into the optimistic regret minimization framework, while maintaining computational efficiency given oracle-access to an FMDP planner. Keeping computational oracle-efficiency is a difficult challenge in factored MDPs and especially hard when structure is unknown, since the number of possible structure configurations is highly exponential. Furthermore, our algorithm easily accommodates any level of structure knowledge, and is therefore extremely useful when additional prior domain knowledge is available. We note that while structure learning in FMDPs was previously studied by Strehl et al. [2007], Diuk et al. [2009], Chakraborty and Stone [2011], Hallak et al. [2015], Guo and Brunskill [2017], none of them provide regret guarantees.

To make our algorithms compatible with existing approximate FMDP planners, we also study FMDPs with non-factored actions. To the best of our knowledge, existing planners require small non-factored action space which is not compatible with the FMDP regret minimization literature. To mitigate this gap, we show that even when the oracle is limited to non-factored actions, a variant of our algorithm can still be implemented efficiently and achieve similar near-optimal regret bounds.

Finally, we leverage the techniques presented in this paper to prove a novel lower bound for regret minimization in FMDPs with known structure. This is the first lower bound to show that the regret must scale exponentially with the scope size $m$, and the first to utilize connections between different factors in a non-trivial way (i.e., with scope size larger than 1). Furthermore, it improves previous lower bounds by a factor of $\sqrt{d}$ and closes the gap to the state-of-the-art regret bound of Chen et al. [2021], thus establishing the minimax optimal regret in this setting.

Our algorithms make oracle use of FMDP planners. However, even where an FMDP can be represented concisely, solving for the optimal policy may take exponentially long in the most general case [Goldsmith et al., 1997, Littman, 1997]. Our focus in this paper is upon the statistical aspect of the learning problem, and we therefore assume oracle-access to an FMDP planner. We emphasize that the oracle assumption appears in all previous regret minimization algorithms. Furthermore, except for the DORL algorithm of Xu and Tewari [2020], all previous algorithms run in time exponential in $d$ even with access to a planning oracle (and known structure). We stress that in many cases of interest, effective approximate planners do exist [Boutilier et al., 2000, Koller and Parr, 2000, Schuurmans and Patrascu, 2001, Guestrin et al., 2001, 2003, Sanner and Boutilier, 2005, Delgado et al., 2011].

## 2 Preliminaries

An infinite-horizon average-reward MDP is described by a tuple $M = (S, A, P, R)$, where $S$ and $A$ are finite state and action spaces, respectively, $P : S \times A \to \Delta_S$ is the transition function[1], and $R : S \times A \to \Delta_{[0,1]}$ is the reward function with expectation $r(s, a) = \mathbb{E}[R(s, a)]$.

The interaction between the MDP and the learner proceeds as follows. The learner starts in an arbitrary initial state $s^1 \in S$. For $t = 1, 2, \ldots$, the learner observes the current state $s^t \in S$, picks an action $a^t \in A$ and earns a reward $r^t$ sampled from $R(s^t, a^t)$. Then, the environment draws the next state $s^{t+1} \sim P(\cdot \mid s^t, a^t)$ and the process continues.

A policy $\pi : S \to A$ is a mapping from states to actions, and its *gain* is defined by the average-reward criterion: $\lambda(M, \pi, s) \stackrel{\text{def}}{=} \lim_{T \to \infty} \frac{1}{T} \mathbb{E}\left[\sum_{t=1}^{T} r(s^t, \pi(s^t)) \mid s^1 = s\right]$, where $s^{t+1} \sim P(\cdot \mid s^t, \pi(s^t))$.

In order to derive non-trivial regret bounds, one must constrain the connectivity of the MDP [Bartlett and Tewari, 2009]. We focus on *communicating* MDPs, i.e., MDPs with finite *diameter $D < \infty$*.

**Definition 1.** Let $T(s' \mid M, \pi, s)$ be the random variable for the first time step in which state $s'$ is reached when playing a stationary policy $\pi$ in an MDP $M$ with initial state $s$. The diameter of $M$ is defined as $D(M) \stackrel{\text{def}}{=} \max_{s \neq s' \in S} \min_{\pi:S \to A} \mathbb{E}[T(s' \mid M, \pi, s)]$.

For communicating MDPs, neither the optimal policy nor its gain depend on the initial state $s^1$. We denote them by $\pi^\star(M) = \arg\max_{\pi:S \to A} \lambda(M, \pi, s^1)$ and $\lambda^\star(M) = \lambda(M, \pi^\star, s^1)$, respectively.

---

[1]$\Delta_X$ denotes the set of distributions over a set $X$.

We measure the performance of the learner by the *regret*. That is, the difference between the expected gain of the optimal policy in $T$ steps and the cumulative reward obtained by the learner up to time $T$, i.e., $\text{Reg}_T(M) \overset{\text{def}}{=} \sum_{t=1}^{T} (\lambda^\star(M) - r^t)$, where $r^t \sim R(s^t, a^t)$ and $a^t$ is chosen by the learner.

## 2.1 Factored MDPs

Factored MDPs inherit the above definitions, but also possess some conditional independence structure that allows compact representation. We follow the factored MDP definition of Osband and Van Roy [2014], which generalizes the original definition of Boutilier et al. [2000], Kearns and Koller [1999] to allow a factored action space as well. We start with a definition of a factored set and scope operation.

**Definition 2.** A set $X$ is called *factored* if it can be written as a product of $n$ sets $X_1, \dots, X_n$, i.e., $X = X_1 \times \cdots \times X_n$. For any subset of indices $Z = \{i_1, \dots, i_{|Z|}\} \subseteq \{1, \dots, n\}$, define the scope set $X[Z] = X_{i_1} \times \cdots \times X_{i_{|Z|}}$. Further, for any $x \in X$ define the scope variable $x[Z] \in X[Z]$ to be the value of the variables $x_i \in X_i$ with indices $i \in Z$. For singleton sets we write $x[i]$ for $x[\{i\}]$.

Next, we define the factored reward and transition functions. We use the notations $X = S \times A$ for the state-action space, $d$ for the number of state factors and $n$ for the number of state-action factors.

**Definition 3.** A reward function $R$ is called factored over $X = X_1 \times \cdots \times X_n$ with scopes $Z_1^r, \dots, Z_\ell^r$ if there exist functions $\{R_j : X[Z_j^r] \to \Delta_{[0,1]}\}_{j=1}^\ell$ with expectations $r_j(x[Z_j^r]) = \mathbb{E}[R_j(x[Z_j^r])]$ such that for all $x \in X$: $R(x) = \frac{1}{\ell} \sum_{j=1}^\ell R_j(x[Z_j^r])$. Note that when a reward $r = \frac{1}{\ell} \sum_{j=1}^\ell r_j$ is sampled from $R(x)$, the learner observes every $r_j$ individually.

**Definition 4.** A transition function $P$ is called factored over $X = X_1 \times \cdots \times X_n$ and $S = S_1 \times \cdots \times S_d$ with scopes $Z_1^P, \dots, Z_d^P$ if there exist functions $\{P_i : X[Z_i^P] \to \Delta_{S_i}\}_{i=1}^d$ such that for all $x \in X$ and $s' \in S$: $P(s' \mid x) = \prod_{i=1}^d P_i(s'[i] \mid x[Z_i^P])$. That is, given a state-action pair $x$, factor $i$ of $s'$ is independent of its other factors, and is determined only by $x[Z_i^P]$.

Thus, a factored MDP (FMDP) is defined by an MDP whose reward and transition functions are both factored, and is fully characterized by the tuple $M = \left( \{X_i\}_{i=1}^n, \{S_i, Z_i^P, P_i\}_{i=1}^d, \{Z_j^r, R_j\}_{j=1}^\ell \right)$. As opposed to previous works [Osband and Van Roy, 2014, Xu and Tewari, 2020, Tian et al., 2020, Chen et al., 2021] that assume known factorization, in this paper the learner does not have any prior knowledge of the scopes $Z_1^P, \dots, Z_d^P$ or $Z_1^r, \dots, Z_\ell^r$, and they need to be learned from experience. However, the learner has a bound $m$ on the size of the scopes, i.e., $|Z_i^P| \le m$ and $|Z_j^r| \le m$ $\forall i, j$. See remarks on unknown scope size and variable scope sizes in Appendix B.

**Remark** (FMDP encoding size). Let the action factorization $A = A_{d+1} \times \cdots \times A_n$, factor size $W = \max\{\max_{1 \le i \le d} |S_i|, \max_{d+1 \le i \le n} |A_i|\}$ and $L = \max_{Z:|Z|=m} |X[Z]|$. The encoding size is $O(dWL + \ell L + (d + \ell)m \log n)$. Importantly, the encoding is only polynomial in $d$ while the number of states $W^d$ is exponential. It is however exponential in the (much smaller) scope size as $L \approx W^m$.

## 3 Structure Learning in FMDPs

In order to keep sample efficiency even when the structure of the FMDP is unknown, the learner must be able to detect the actual scopes $Z_1^P, \dots, Z_d^P$ and $Z_1^r, \dots, Z_\ell^r$. Let's focus on learning the scopes for the transition function first, as the technique for the reward function is similar. Our structure learning approach is based on a simple yet powerful observation by Strehl et al. [2007]. Since the $i$-th factor of the next state depends only on the scope $Z_i^P$, an empirical estimate of $P_i$ should remain relatively similar whether it is computed using $Z_i^P$ or $Z_i^P \cup Z$ for any other scope $Z \subseteq \{1, \dots, n\}$.

Formally, define the empirical transition function for factor $i$ based on scope $Z$ at time step $t$ as $\bar{P}_{i,Z}^t(w \mid v) = \frac{N_{i,Z}^t(v,w)}{\max\{N_Z^t(v), 1\}}$ for every $(v, w) \in X[Z] \times S_i$, where $N_Z^t(v)$ is the number of times we have visited a state-action pair $x$ such that $x[Z] = v$ up to time step $t$, and $N_{i,Z}^t(v, w)$ is the number of times this visit was followed by a transition to a state $s'$ such that $s'[i] = w$. Regardless of the additional scope $Z$, the expected value of $\bar{P}_{i,Z_i^P \cup Z}^t(s'[i] \mid x[Z_i^P \cup Z])$ remains $P_i(s'[i] \mid x[Z_i^P])$.

We leverage this observation to define *consistent* scopes. A scope $Z$ of size $m$ is consistent for factor $i$ if for every other scope $Z'$ of size $m$, $v \in X[Z \cup Z']$ and $w \in S_i$,

$$\left| \bar{P}^t_{i,Z \cup Z'}(w|v) - \bar{P}^t_{i,Z}(w|v[Z]) \right| \le 2 \cdot \epsilon^t_{i,Z \cup Z'}(w|v), \tag{1}$$

where $\epsilon^t_{i,Z}(w \mid v) \overset{\text{def}}{=} \sqrt{\frac{18 \bar{P}^t_{i,Z}(w|v)\tau^t}{\max\{N^t_Z(v),1\}}} + \frac{18\tau^t}{\max\{N^t_Z(v),1\}}$ is the radius of the confidence set, $\tau^t = \log(6dWLt/\delta)$ is a logarithmic factor and $\delta$ is the confidence parameter.

There are two important properties that hold by a simple application of Hoeffding inequality. First, the actual scope $Z^P_i$ will always be consistent with high probability. Second, if a different scope $Z$ is consistent, then the empirical estimates $\bar{P}^t_{i,Z^P_i}$ and $\bar{P}^t_{i,Z}$ must be close, since both are close to $\bar{P}^t_{i,Z^P_i \cup Z} = \bar{P}^t_{i,Z \cup Z^P_i}$. Therefore, they are close to the true transition function $P_i$ with high probability.

Thus, our approach for structure learning is to eliminate inconsistent scopes. In the next section we show how this idea can be combined with the method of *optimism in face of uncertainty* for regret minimization in FMDPs. This approach works similarly for learning the scopes of the reward function. Formally, define the empirical reward function for reward factor $j$ based on scope $Z$ at time $t$ as $\bar{r}^t_{j,Z}(v) = \frac{1}{\max\{N^t_Z(v),1\}} \sum_{h=1}^{t-1} r^h_j \cdot \mathbb{I}\{(s^h, a^h)[Z] = v\}$ for every $v \in X[Z]$, where $\mathbb{I}\{\cdot\}$ is the indicator. Similarly to the transitions, a scope $Z$ of size $m$ is *reward consistent* for reward factor $j$ if for every other scope $Z'$ of size $m$ and $v \in X[Z \cup Z']$,

$$\left| \bar{r}^t_{j,Z \cup Z'}(v) - \bar{r}^t_{j,Z}(v[Z]) \right| \le 2 \cdot \epsilon^t_{Z \cup Z'}(v) \overset{\text{def}}{=} 2 \cdot \sqrt{18\tau^t / \max\{N^t_{Z \cup Z'}(v),1\}}.$$

## 4  The SLF-UCRL Algorithm

Our algorithm Structure Learn Factored UCRL (SLF-UCRL) follows the known framework of optimism in face of uncertainty while learning the structure of the FMDP. A sketch is given in Algorithm 1 and the full algorithm can be found in Appendix A. Similarly to the UCRL2 algorithm [Jaksch et al., 2010], we split the time into episodes. In the beginning of every episode we compute an optimistic policy and play it for the entire episode. The episode ends once the number of visits to some $v \in X[Z \cup Z']$ is doubled, where $Z \ne Z'$ are two scopes of size $m$. That is, the number of times we visited a state-action pair $x$ with $x[Z \cup Z'] = v$ is doubled. Note that the standard doubling technique of Jaksch et al. [2010], i.e., when the number of visits to some state-action pair is doubled, will result in regret that depends polynomially on the size of the state-action space, which is exponentially larger than the size of its factors. Moreover, our doubling scheme is different than Xu and Tewari [2020], where the episode size grows arithmetically. This allows us to obtain tighter regret bound that depends on the different sizes of all the factors, and not just the biggest one $L \approx W^m$.

While optimism is a standard framework for regret minimization, our algorithm features two novel techniques to handle unknown structure. First, we show how structure learning can be combined with optimism through the concept of consistent scopes. This already gives an algorithm with bounded regret, but requires exponential running time and space complexity. Second, in Sections 4.1 and 4.2 we present a novel construction that allows to compute the optimistic policy in an oracle-efficient and space efficient manner, although the number of consistent factored structures is clearly exponential.

For every factor $i$ we maintain a set $\widetilde{\mathcal{Z}}^k_i$ of its consistent scopes up to episode $k$ (we keep a similar set $\widetilde{\mathcal{R}}^k_j$ for every reward factor $j$). In the beginning of the episode we construct an optimistic MDP $\widetilde{M}^k$ out of all possible configurations of consistent scopes. We then compute the optimal policy $\tilde{\pi}^k$ of $\widetilde{M}^k$, extract the optimistic policy $\pi^k$ and play it throughout the episode. In what follows, we denote by $t_k$ the first time step of episode $k$, and slightly abuse notation by using $\bar{P}^k, \epsilon^k, N^k$ for $\bar{P}^{t_k}, \epsilon^{t_k}, N^{t_k}$.

**Remark** (Computational complexity). The computational complexity of our algorithm scales exponentially with the scope size $m$, but polynomially with the number of factors $n, d, \ell$. This dependence is unavoidable [Abbeel et al., 2006, Strehl et al., 2007] since the number of possible scopes is $\binom{n}{m}$ and the size of the FMDP encoding is also exponential in $m$. In fact, the complexity of all previous regret minimization algorithms (except Xu and Tewari [2020]) is exponential even in the number of factors $d$ and not just in the scope size $m$. Since FMDPs with large scope size are not practical (their representation is huge), one should think of $m$ as very small compared to $n, d, \ell$.

---

**Algorithm 1** SLF-UCRL Sketch

---

**Input:** $\delta, m, S = \{S_i\}_{i=1}^d, S \times A = X = \{X_i\}_{i=1}^n$.
Initialize visit counters and sets of consistent scopes.
**for** $k = 1, 2, \ldots$ **do**
    Start new episode $k$, and compute empirical transition function $\bar{P}^k$ and confidence bounds $\epsilon^k$.
    Eliminate inconsistent scopes (Algorithm 2), and construct optimistic MDP $\widetilde{M}^k$.
    Compute optimal policy $\tilde{\pi}^k$ of $\widetilde{M}^k$ using oracle, and extract optimistic policy $\pi^k$.
    Execute policy $\pi^k$ until there are scopes $Z \neq Z'$ of size $m$ and $v \in X[Z \cup Z']$ such that the
    number of visits to some state-action pairs $x$ with $x[Z \cup Z'] = v$, is doubled.
**end for**

---

**Algorithm 2** Eliminate Inconsistent Scopes Sketch

---

**for** $i = 1, \ldots, d$ and $Z \in \widetilde{\mathcal{Z}}_i^{k-1}$ **do**
    **for** $Z' \subseteq \{1, \ldots, n\}$ of size $m$ and $v \in X[Z \cup Z']$ and $w \in S_i$ **do**
      **if** $|\bar{P}_{i,Z\cup Z'}^k(w \mid v) - \bar{P}_{i,Z}^k(w \mid v[Z])| > 2 \cdot \epsilon_{i,Z\cup Z'}^k(w \mid v)$ **then**
        Eliminate inconsistent scope: $\widetilde{\mathcal{Z}}_i^k \leftarrow \widetilde{\mathcal{Z}}_i^k \setminus \{Z\}$, and BREAK.
      **end if**
    **end for**
**end for**
\# Inconsistent reward scopes are eliminated from $\widetilde{\mathcal{R}}_j^k$ similarly, for every $j = 1, \ldots, \ell$.

---

**Remark** (Partial structure knowledge). The SLF-UCRL algorithm easily accommodates any level of knowledge regarding the structure. That is, the consistent scopes sets can be adjusted if some scopes are known or have a known compact representation (e.g., decision trees). The algorithm's complexity and regret scale naturally with the level of structure knowledge, making it extremely useful when specific domain knowledge is available (e.g., dynamics of some physical systems in robotics).

## 4.1 Constructing the Optimistic MDP $\widetilde{M}^k$

As our construction generalizes the one of Xu and Tewari [2020] to the case of unknown structure, we start with a brief overview of their method. With known structure, their optimistic MDP keeps the same state space $S$ but has an extended action space $A \times S$, where playing action $(a, s')$ in state $s$ corresponds to playing action $a$ and using a transition function that puts all the uncertainty in the direction of state $s'$, such that for each factor $i$ the $L_1$ distance between the empirical and optimistic transition functions is bounded by $\sum_{w \in S_i} \epsilon_{i,Z_i^P}^k(w \mid (s,a)[Z_i^P]) = \widetilde{O}\big(\sqrt{|S_i|/N_{Z_i^P}^k\big((s,a)[Z_i^P]\big)}\big)$.

Formally, let $\mathcal{W}_{i,Z}^k(w \mid v) = \min\{\epsilon_{i,Z}^k(w \mid v), \bar{P}_{i,Z}^k(w \mid v)\}$ and then the probability that in the optimistic MDP the $i$-th factor of the next state is $w$ after playing $(a, s')$ in state $s$ is

$$\bar{P}_{i,Z_i^P}^k\big(w \mid (s,a)[Z_i^P]\big) - \mathcal{W}_{i,Z_i^P}^k\big(w \mid (s,a)[Z_i^P]\big) + \mathbb{I}\{w = s'[i]\} \cdot \sum_{w' \in S_i} \mathcal{W}_{i,Z_i^P}^k\big(w' \mid (s,a)[Z_i^P]\big).$$

The $j$-th reward factor of this action is the empirical estimate plus an additional optimistic bonus, i.e.,

$$\min\Big\{1, \bar{r}_{j,Z_j^r}^k\big((s,a)[Z_j^r]\big) + \epsilon_{Z_j^r}^k\big((s,a)[Z_j^r]\big)\Big\}.$$

Notice that this optimistic MDP is factored, that the number of state-action factors increased by $d$, and that the scope size increased by only $1$. Thus, this method indeed keeps oracle-efficiency.

The naive way to extend this idea to unknown structure is to compute the optimistic MDP for every configuration of consistent scopes, and pick the most optimistic one, i.e., the configuration in which the optimal gain is the biggest. However, this requires exponential number of calls to the oracle.

Instead, we propose to extend the action space even further so the policy can pick the scopes as well as the actions. That is, the extended action space is $\widetilde{A}^k = A \times S \times \widetilde{\mathcal{Z}}_1^k \times \cdots \times \widetilde{\mathcal{Z}}_d^k \times \widetilde{\mathcal{R}}_1^k \times \cdots \times \widetilde{\mathcal{R}}_\ell^k$, and playing action $\tilde{a} = (a, s', Z_1, \ldots, Z_d, z_1, \ldots, z_\ell)$ in state $s$ corresponds to playing action $a$, using a reward function according to scopes $z_1, \ldots, z_\ell$, and using a transition function according to

scopes $Z_1, \ldots, Z_d$ that puts all the uncertainty in the direction of $s'$. Formally, for every reward factor $j$ define $\tilde{r}_j^k(\tilde{x}) = \min\{1, \bar{r}_{j,z_j}^k((s,a)[z_j]) + \epsilon_{z_j}^k((s,a)[z_j])\}$, where $\tilde{x} = (s, \tilde{a})$. For every factor $i$ and $w \in S_i$ define

$$
\begin{aligned}
\widetilde{P}_i^k(w|\tilde{x}) &\overset{\text{def}}{=} \bar{P}_{i,Z_i}^k(w \mid (s,a)[Z_i]) - \mathcal{W}_{i,Z_i}^k(w \mid (s,a)[Z_i]) \\
&\quad + \mathbb{I}\{w = s'[i]\} \cdot \sum_{w' \in S_i} \mathcal{W}_{i,Z_i}^k(w' \mid (s,a)[Z_i]).
\end{aligned} \tag{2}
$$

Unfortunately, although this elegant construction looks like a factored MDP, it is in fact not factored. Specifically, the transition and reward functions become non-factored because each factor can now depend on all the factors of the state-action space (this is determined by the policy choosing the scopes), i.e., the scope size is now $n$. Nevertheless, in the following section we show that the optimal policy of this optimistic MDP can still be computed by the oracle. To that end, we construct a slightly larger MDP that has the same optimal policy and gain, while being factored with small scopes.

## 4.2 From Optimistic MDP $\widetilde{M}^k$ to Optimistic Factored MDP $\widehat{M}^k$

We construct a factored MDP $\widehat{M}^k$ that simulates exactly the dynamics of the optimistic MDP $\widetilde{M}^k$. The idea is to stretch each time step to $2 + \log n$ steps. In the first step the policy chooses a combined action $\tilde{a}$ as described in Section 4.1, in the next $\log n$ steps relevant factors are extracted according to the policy's choices, and in the last step the transition is performed according to $\widetilde{P}^k$ (Eq. (2)). Since the relevant factors for the transition were already extracted, this time the scope size remains small.

For $\widehat{M}^k$, we keep the extended action space $\widehat{A}^k = \widetilde{A}^k$ and extend the state space $\widehat{S}^k$ to contain the state, steps counter, the policy's picked scopes and optimistic assignment state, and a "temporary" work space. Formally, $\widehat{S}^k = S \times \{0, 1, \ldots, \log n + 1\} \times S \times \widetilde{\mathcal{Z}}_1^k \times \cdots \times \widetilde{\mathcal{Z}}_d^k \times \widetilde{\mathcal{R}}_1^k \times \cdots \times \widetilde{\mathcal{R}}_\ell^k \times \Omega^{(d+\ell)m}$, where $S$ keeps the state, $\{0, 1, \ldots, \log n + 1\}$ is a counter of the current step within the actual time step, and $S \times \widetilde{\mathcal{Z}}_1^k \times \cdots \times \widetilde{\mathcal{Z}}_d^k \times \widetilde{\mathcal{R}}_1^k \times \cdots \times \widetilde{\mathcal{R}}_\ell^k$ keeps the policy's picked scopes and optimistic assignment state. For each factor $i$ (also for each reward factor $j$) and index $e \in \{1, \ldots, m\}$, we have a separate "temporary" work space $\Omega = \omega^n \times \omega^{n/2} \times \cdots \times \omega^2 \times \omega$ that allows extracting the $e$-th element of the scope for the transition of factor $i$ while maintaining small scope sizes. Here, $\omega = (\bigcup_{i=1}^d S_i) \cup (\bigcup_{i=d+1}^n A_i)$ keeps one factor (state or action), so $|\omega| = W$.

A state $s$ in $M$ is mapped to $(s, 0, \perp)^2$ and taking action $\tilde{a} = (a, s', Z_1, \ldots, Z_d, z_1, \ldots, z_\ell)$ results in a deterministic transition to $(s, 1, s', Z_1, \ldots, Z_d, z_1, \ldots, z_\ell, \tau)$, where $\tau = (\tau_{i,e}) \in \Omega^{(d+\ell)m}$. The state-action pair $(s, a)$ is copied to each of the work spaces, i.e., $\tau_{i,e} = (s, a, \perp)$. The next $\log n$ steps are used to extract the relevant scopes. The policy has no effect in these steps since $a, s'$ and the chosen scopes are now encoded into the state. In these $\log n$ steps, for each $(i, e)$, we eliminate half of $\tau_{i,e}$ in each step according to its chosen scope $Z_i$, until we are left with the $e$-th factor of $(s, a)[Z_i]$. The elimination steps require scopes of size only $4$ since each factor of the next step needs to choose between two factors from the previous step (while considering the scope $Z_i$ chosen by the policy and the counter). The final step performs the transition according to $\widetilde{P}^k$, but notice that now it only requires scopes of size $m + 3$. The reason is that now $(s, a)[Z_i]$ has a fixed location within the state, i.e., $(s, a)[Z_i][e]$ is in the last factor of $\tau_{i,e}$ (in addition, the counter, $s'[i]$ and $s[i]$ need to be taken into consideration). At this point the agent also gets the reward $\tilde{r}^k$, whereas the reward in all other time steps is $0$. Similarly to the transitions, the reward scopes are of size $m + 1$ because $(s, a)[z_j]$ has a fixed location (and the counter should also be considered). For more details see Appendix A.

It is easy to see that $\widehat{M}^k$ simulates $\widetilde{M}^k$ exactly, because every $2 + \log n$ steps are equivalent to one step in $\widetilde{M}^k$. In terms of computational complexity, any planner that is able to solve $M$ can also solve $\widehat{M}^k$, since it is factored and polynomial in size when compared to $M$. Indeed, the scope size is $m + 3$ (compared to $m$), the number of state factors is $3d + \ell + 1 + 2nm(d + \ell)$ (compared to $d$), the number of action factors is $n + d + \ell$ (compared to $n - d$), the size of each state factor is bounded by $\max\{W, \binom{n}{m}\}$ (compared to $W$), and finally the size $L$ is replaced by $\max\{L, \binom{n}{m}\}W^2(2 + \log n)$. Given the optimal policy $\hat{\pi}^k$ for $\widehat{M}^k$, we can easily extract the optimal policy $\tilde{\pi}^k(s) = \hat{\pi}^k((s, 0, \perp))$ for $\widetilde{M}^k$, and the optimistic policy $\pi^k(s) = \tilde{\pi}^k(s)[1] = \hat{\pi}^k((s, 0, \perp))[1]$ for the original MDP $M$.

---

[2]We use $\perp$ to indicate that the rest of the state is irrelevant.

## 4.3 Avoiding Large Factors

One shortcoming of the above construction is that the factor size may be significantly larger, i.e., $\binom{n}{m}$ instead of $W$ in the original FMDP. As mentioned before, $m$ is considered to be small, and yet one might prefer to keep the factor size small at the expense of adding a few extra factors and increasing the reward scope size by $1$. In what follows, we show that this is indeed possible because each action factor we added for choosing a consistent scope is already factored internally into $m$ factors of size $n$.

We view the extended action space as $A \times S \times \{1, \ldots, n\}^{m(d+\ell)}$ which has $n + m(d+\ell)$ factors of size $\max\{W, n\}$. Similarly, we can view the state space as $2d + 1 + m(d+\ell) + 2nm(d+\ell)$ factors of the same size. Luckily we can still keep the same $m + 3$ scope size, since the consistent scopes are used only in the $\log n$ intermediate steps in which the scope size was $4$ and now becomes $m + 3$. However, this gives rise to a new problem: now the policy might choose inconsistent scopes because the action space is not restricted to consistent scopes anymore. To overcome this issue, we enforce the optimal policy in $\widehat{M}^k$ to use only consistent scopes by adding $2(d+\ell)$ binary factors. These factors make sure that any policy that uses an inconsistent scope will never earn a reward.

All the new binary factors start as $1$, and we refer to them as bits. When the counter is $0$, the $i$-th bit becomes $0$ if the chosen scope for factor $i$ is inconsistent. Similarly, the $(d+j)$-th bit checks the chosen scope for reward factor $j$. This requires them to have scope size $m + 2$, and in the next $\log(d+\ell)$ steps we extract out of them one bit that says if an inconsistent scope was chosen. This is done similarly to the extraction of relevant scopes and requires the counter to reach $\max\{\log(d+\ell), \log n\} + 1$ instead of $\log n + 1$. Finally, when giving a reward in the last step, the reward function also considers the extracted bit and gives $0$ reward if it is $0$. Since it cannot turn back to $1$, this bit ensures that a policy that uses an inconsistent scope has a gain of $0$.

## 4.4 Regret Analysis

In Appendix B we prove the following regret bound for SLF-UCRL. Here we review the main ideas.

**Theorem 1.** *Running SLF-UCRL on a factored MDP with unknown structure ensures, with probability at least $1 - \delta$,*

$$Reg_T(M) = \widetilde{O}\left(\sum_{i=1}^{d} \sum_{Z:|Z|=m} D\sqrt{|S_i||X[Z_i^P \cup Z]|T} + \frac{1}{\ell}\sum_{j=1}^{\ell} \sum_{Z:|Z|=m} \sqrt{|X[Z_j^r \cup Z]|T}\right).$$

In the worst-case regret, this regret bound becomes $Reg_T(M) = \widetilde{O}(\binom{n}{m}dD\sqrt{WL^2T})$. In comparison to the regret bound of Xu and Tewari [2020] for the known structure case, our bound is worse by only a factor of $\binom{n}{m}\sqrt{L}$. While the exponential dependence in $m$ (hidden already in $L$) is unavoidable, it is an important open problem whether the multiplicative dependence in $\binom{n}{m}$ is necessary (note that it directly stems from the level of structure knowledge and may be much smaller with some domain knowledge). We believe that the $\sqrt{L}$ factor can be avoided with methods such as the meteorologist algorithm of Diuk et al. [2009], since it comes from our simple structure learning method, i.e., comparing all pairs of scopes $Z \neq Z'$ of size $m$. Still, it is highly unclear how to incorporate these methods in a regret minimization algorithm. Finally, we stress that ignoring the unknown factored structure leads to regret polynomial in the number of states, which is exponential compared to ours.

*Proof sketch.* Regret analysis for optimistic algorithms has two main parts: (1) *optimism* - show that the optimal gain in the optimistic model $\widetilde{M}^k$ is at least as large as $\lambda^\star(M)$ for all episodes $k$ with high probability; (2) *deviation* - bound the difference between the optimistic policy's gains in $M$ and $\widetilde{M}^k$.

Optimism follows directly from the consistent scopes definition and standard concentration inequalities. Specifically, since the true scopes are always consistent with high probability, the optimistic policy in the optimistic model maximizes its gain while choosing scopes from a set that contains the true scopes. For the deviation, we need to bound the distance between the true and optimistic dynamics along the trajectory visited in each episode $k$. That is, we need to relate $\Delta_t = \|\widetilde{P}^k(\cdot|\tilde{x}^t) - P(\cdot|x^t)\|_1$ to the confidence radius $\epsilon^k$, where $\tilde{\pi}^k(s^t) = (a^t, s'^t, Z_1^t, \ldots, Z_d^t, z_1^t, \ldots, z_\ell^t)$, $x^t = (s^t, a^t)$ and $\tilde{x}^t = (s^t, \tilde{\pi}^k(s^t))$. Then, we can sum the confidence radii over $t = 1, \ldots, T$ and get the final bound.

To that end, we utilize the transition factorization to bound $\Delta_t \leq \sum_{i=1}^{d} \left\| \widetilde{P}_i^k(\cdot|\tilde{x}^t) - P_i(\cdot|x^t[Z_i^P]) \right\|_1$. Then, for each $i$ we can use the definition of the optimistic transition function $\widetilde{P}^k$ (Eq. (2)) to get

$$\Delta_t \lesssim \sum_{i=1}^{d} \left\| \bar{P}_{i,Z_i^P}^k(\cdot \mid x^t[Z_i^P]) - P_i(\cdot \mid x^t[Z_i^P]) \right\|_1 + \sum_{i=1}^{d} \left\| \bar{P}_{i,Z_i^t}^k(\cdot \mid x^t[Z_i^t]) - \bar{P}_{i,Z_i^P}^k(\cdot \mid x^t[Z_i^P]) \right\|_1.$$

The first term measures the difference between the empirical and true dynamics on the correct scopes $Z_i^P$ and can therefore be bounded with standard concentration inequalities. For the second term we utilize the fact that the chosen scopes $Z_i^t$ must be consistent. Therefore, we can bound it using Eq. (1) by $\approx \sum_{i=1}^{d} \sum_{w \in S_i} \epsilon_{i,Z_i^P \cup Z_i^t}^k(w \mid x^t[Z_i^P \cup Z_i^t]) \lesssim \sum_{i=1}^{d} \sqrt{|S_i| / N_{Z_i^P \cup Z_i^t}^k(x^t[Z_i^P \cup Z_i^t])}$. $\qquad \square$

## 5 Factored MDPs with Non-Factored Actions

So far we assumed that both the state and action spaces are factored. While this model is very general, it also requires an oracle that can solve it. However, almost all existing approximate FMDP planners do not address factored action spaces. Moreover, implicitly they assume that the action set is small (or with very unique structure), as they pick a greedy policy with respect to some Q-function estimation.

To make our algorithm more compatible with approximate planners, in this section we do not assume that the action space is factored, and our oracle is limited to such FMDPs. We show that a variant of our algorithm can still achieve similar regret bounds and maintain computational efficiency. This makes our algorithm much more practical than the DORL algorithm of Xu and Tewari [2020]. The FMDP definition we adopt assumes that the state space is factored $S = S_1 \times \cdots \times S_d$, and that the transition function is factored, only with respect to the state space, in the following manner. The factored reward function is defined similarly, but to simplify presentation, we assume it is known.

**Definition 5.** Transition function $P$ is called factored over $S = S_1 \times \cdots \times S_d$ with scopes $Z_1^P, \ldots, Z_d^P$ if there exist functions $\{P_i : S[Z_i^P] \times A \to \Delta_{S_i}\}_{i=1}^{d}$ s.t. $P(s' \mid s, a) = \prod_{i=1}^{d} P_i(s'[i] \mid s[Z_i^P], a)$.

We focus on known structure to convey the main ideas, but in Appendix E we show that the methods presented here can be extended to handle unknown structure. The DORL algorithm [Xu and Tewari, 2020] highly relies on the factored action space, because the optimistic MDP is defined using the huge (yet factored) action space $A \times S$ that allows incorporating an optimistic estimate of the dynamics. Instead, we propose to spread the transition across $2 + d$ steps. In the first step the policy picks an action, step $i + 1$ performs the $i$-th factor optimistic transition, and the last step completes the move.

Formally, the state space of $\widetilde{M}^k$ is $\widetilde{S} = S \times \{0, 1, \ldots, d+1\} \times A \times S \times \{0, 1\}$, where $S$ keeps the state, $\{0, 1, \ldots, d+1\}$ is a counter of the current step within the actual time step, $A$ keeps the policy's chosen action, another $S$ helps perform the transition, and the last bit validates that the chosen actions are legal. The action space of $\widetilde{M}^k$ is $\widetilde{A} = A \cup (\bigcup_{i=1}^{d} S_i)$. The size of $\widetilde{A}$ is $\max\{|A|, W\}$ which is exponentially smaller compared to $|A| W^d$ in the original construction of Xu and Tewari [2020].

A state $s$ in $M$ is mapped to $(s, 0, \perp)$ and action $a \in A$ deterministically transitions to $(s, 1, a, \perp)$, while the other actions are not legal at this state. Picking an illegal action turns the last bit to 0 (it starts as 1), canceling all rewards similarly to Section 4.3. In state $(s, i, a, w_1, \ldots, w_{i-1}, \perp)$, legal actions are $S_i$, and action $w \in S_i$ transitions to state $(s, i+1, a, w_1, \ldots, w_{i-1}, w_i, \perp)$ with probability

$$\bar{P}_{i,Z_i^P}^k(w_i \mid s[Z_i^P], a) - \mathcal{W}_{i,Z_i^P}^k(w_i \mid s[Z_i^P], a) + \mathbb{I}\{w_i = w\} \cdot \sum_{w' \in S_i} \mathcal{W}_{i,Z_i^P}^k(w' \mid s[Z_i^P], a).$$

Finally, $(s, d+1, a, w_1, \ldots, w_d, b)$ transitions deterministically to $(s', 0, \perp)$ for $s' = (w_1, \ldots, w_d)$.

Similarly to Section 4.1, one can see that the scope size is now $m + 3$, the number of factors is $2d + 3$, the size of each factor is bounded by $\max\{W, |A|, d+2\}$, and that the number of actions remains small. Thus we can use the limited oracle in order to solve the optimistic MDP. As for the regret bound, it is easy to verify that optimism still holds, but it is not clear that we can still bound the deviation because now the policy in the optimistic model has significantly more "power" – it chooses the uncertainty direction for factor $i$ after the realizations for factors $1, \ldots, i-1$ of the next state are already revealed. Next, we show that it can still be bounded similarly since the actual action of the policy is chosen before the realizations are revealed (the action is chosen in the first of $d+1$ steps).

To see that, consider an MDP $M'$ that models the exact same process as $M$ but resembles our optimistic MDP as each time step is stretched over $d + 2$ steps. The state space of $M'$ is $\widetilde{S}$ like $\widetilde{M}^k$, and taking action $a \in A$ in state $(s, 0, \perp)$ transitions to state $(s, 1, a, \perp)$. Then, the policy has no effect for $d + 1$ steps and the action is embedded in the state. For every $i$ and $w_i \in S_i$, the probability of transitioning from $(s, i, a, w_1, \ldots, w_{i-1}, \perp)$ to $(s, i+1, a, w_1, \ldots, w_{i-1}, w_i, \perp)$ is simply $P_i(w_i \mid s[Z_i^P], a)$, and finally, $(s, i, a, w_1, \ldots, w_d, b)$ transitions to $(w_1, \ldots, w_d, 0, \perp)$.

Clearly, playing policy $\pi$ in $M$ is equivalent to playing policy $\pi'$ in $M'$ such that $\pi'((s, 0, \perp)) = \pi(s)$. Therefore, $\lambda^\star(M') = \frac{\lambda^\star(M)}{d+2}$ and we can analyze the regret in $M'$ to obtain a similar regret bound to Xu and Tewari [2020]. The full algorithm which we call Non-Factored Actions DORL (NFA-DORL) is found in Appendix C and the full proof of the following regret bound is found in Appendix D.

**Theorem 2.** *Running NFA-DORL on a factored MDP with non-factored actions and known structure ensures with probability $1 - \delta$, $Reg_T(M) = \widetilde{O}(\sum_i D\sqrt{|S_i||S[Z_i^P]||A|T} + \frac{1}{\ell}\sum_j \sqrt{|S[Z_j^r]||A|T})$.*

## 6 Lower Bound

In Appendix F we prove the following lower bound for regret minimization in factored MDPs.

**Theorem 3.** *Let $d > m > 0$. For any algorithm there exists an FMDP with $3d + \log d$ state factors of size at most $\max\{W + 1, \log d + 2\}$, non-factored action space of size $|A|$, and scope size $1 + \max\{m, \log d\}$, such that $\mathbb{E}[Reg_T(M)] = \Omega\left(\sqrt{\frac{d}{\log d}}W^m|A|T\right)$.*

The proof leverages our techniques (e.g., propagating rewards through multi-step factored transitions) in order to embed $dW^m$ multi-arm bandit (MAB) problems into a factored MDP, and make sure that they must be solved sequentially and not in parallel. It features a sophisticated construction to utilize connections between factors in such a way that in each step the learner gets information on just a single MAB, forcing her to solve all the MABs one by one. Our construction is also the first to feature arbitrary scope size $m$, while previous constructions simply take $d$ unrelated factors with scope size 1. As a result, our construction highlights the unique hardness that factored structure might introduce.

This is the first lower bound to show that the regret must scale exponentially with the scope size $m$. Moreover, it improves on previous lower bounds [Tian et al., 2020, Chen et al., 2021] by a factor of $\sqrt{d}$, and matches the state-of-the-art regret bound of Chen et al. [2021] in the known structure case. Thus, our lower bound is tight, proving that this is indeed the minimax optimal regret for FMDPs with known structure. Yet, two intriguing question are left open. First, the optimal regret algorithm of Chen et al. [2021] runs in exponential time, and achieving the same regret with an oracle-efficient algorithm seems like a difficult challenge. Second, extending our lower bound to the unknown structure case is another challenging future direction that can advance us towards discovering whether unknown structure indeed introduces additional hardness in terms of optimal regret.

## 7 Experiments

We test our algorithm on the *SysAdmin* domain [Guestrin et al., 2003] – a network of servers connected by some topology, where failing servers affect the probability of their neighbors to fail and the admin chooses which server to reboot at each time step. Our experiments show that the performance of SLF-UCRL is comparable to that of Factored-UCRL [Osband and Van Roy, 2014] that knows the factored structure in advance, and significantly better than the performance of UCRL [Jaksch et al., 2010] that completely ignores the factorization. Figure 1 shows that,

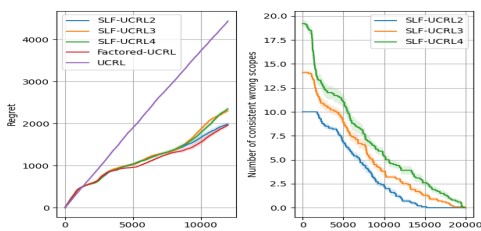

Figure 1: SLF-UCRL performance on circular topology *SysAdmin* with 4 state factors.

for circular topology with 4 servers (i.e., 4 state factors and scope size 3), SLF-UCRL eliminates the wrong scopes (right figure), and has similar regret to Factored-UCRL (left figure). "SLF-UCRL$i$" refers to $i$ factors whose scope needs to be learned, so SLF-UCRL4 has no knowledge of the structure.

For implementation details and more experiments on different topologies and sizes, see Appendix G.

## Acknowledgements

This project has received funding from the European Research Council (ERC) under the European Union'sHorizon 2020 research and innovation program (grant agreement No. 882396), by the Israel Science Foundation(grant number 993/17), Tel Aviv University Center for AI and Data Science (TAD), and the Yandex Initiative for Machine Learning at Tel Aviv University

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
