# A  The SLF-UCRL Algorithm

---

**Algorithm 3** SLF-UCRL

---

**Input:** confidence parameter $\delta$, scope size $m$, state space $S = \{S_i\}_{i=1}^d$, state-action space $S \times A = X = \{X_i\}_{i=1}^n$.

**# Initialization**

Initialize sets of consistent scopes: $\widetilde{\mathcal{R}}_1^0 \leftarrow \cdots \leftarrow \widetilde{\mathcal{R}}_\ell^0 \leftarrow \widetilde{\mathcal{Z}}_1^0 \leftarrow \cdots \leftarrow \widetilde{\mathcal{Z}}_d^0 \leftarrow \{Z \subseteq \{1, \ldots, n\} \mid |Z| = m\}$.

Initialize total visit counters $N$, in-episode visit counters $\nu$ and reward summation variables $r$:

**for** $Z \subseteq \{1, \ldots, n\}$ such that $m \leq |Z| \leq 2m$, $v \in X[Z]$, $j = 1, \ldots, \ell$, $i = 1, \ldots, d$, $w \in S_i$ **do**

$\quad r_{j,Z}(v) \leftarrow N_{i,Z}^0(v, w) \leftarrow \nu_{i,Z}^0(v, w) \leftarrow N_Z^0(v) \leftarrow \nu_Z^0(v) \leftarrow 0$.

**end for**

Initialize time steps counter: $t \leftarrow 1$, and observe initial state $s^1$.

**for** $k = 1, 2, \ldots$ **do**

$\quad$ **# Start New Episode**

$\quad$ Set episode starting time: $t_k \leftarrow t$.

$\quad$ Initialize sets of consistent scopes: $\widetilde{\mathcal{Z}}_i^k \leftarrow \widetilde{\mathcal{Z}}_i^{k-1} \; \forall i$ and $\widetilde{\mathcal{R}}_j^k \leftarrow \widetilde{\mathcal{R}}_j^{k-1} \; \forall j$.

$\quad$ **for** $Z \subseteq \{1, \ldots, n\}$ such that $m \leq |Z| \leq 2m$ and $v \in X[Z]$ **do**

$\quad\quad$ Update visit counters: $\nu_Z^k(v) \leftarrow 0$, $N_Z^k(v) \leftarrow N_Z^{k-1}(v) + \nu_Z^{k-1}(v)$.

$\quad\quad$ **for** $i = 1, \ldots, d$ and $w \in S_i$ **do**

$\quad\quad\quad$ Update visit counters: $\nu_{i,Z}^k(v, w) \leftarrow 0$, $N_{i,Z}^k(v, w) \leftarrow N_{i,Z}^{k-1}(v, w) + \nu_{i,Z}^{k-1}(v, w)$.

$\quad\quad\quad$ Compute empirical transition and reward functions:

$$\bar{P}_{i,Z}^k(w \mid v) = \frac{N_{i,Z}^k(v, w)}{\max\{N_Z^k(v), 1\}} \quad ; \quad \bar{r}_{j,Z}^k(v) = \frac{r_{j,Z}(v)}{\max\{N_Z^k(v), 1\}}.$$

$\quad\quad\quad$ Set confidence bounds:

$$\epsilon_{i,Z}^k(w \mid v) = \sqrt{\frac{18 \bar{P}_{i,Z}^k(w \mid v) \log \frac{6dWLt_k}{\delta}}{\max\{N_Z^k(v), 1\}}} + \frac{18 \log \frac{6dWLt_k}{\delta}}{\max\{N_Z^k(v), 1\}}$$

$$\epsilon_Z^k(v) = \sqrt{\frac{18 \log \frac{6dWLt_k}{\delta}}{\max\{N_Z^k(v), 1\}}}$$

$$\mathcal{W}_{i,Z}^k(w \mid v) = \min\{\epsilon_{i,Z}^k(w \mid v), \bar{P}_{i,Z}^k(w \mid v)\}.$$

$\quad\quad$ **end for**

$\quad$ **end for**

$\quad$ Eliminate inconsistent scopes (Algorithm 4).

$\quad$ Construct optimistic MDP $\widetilde{M}^k$ and compute optimistic policy $\pi^k$ (Algorithm 5).

$\quad$ **# Execute Policy**

$\quad$ **while** $\nu_Z^k((s^t, \pi^k(s^t))[Z]) < N_Z^k((s^t, \pi^k(s^t))[Z]) \; \forall Z \subseteq \{1, \ldots, n\}$ s.t. $m \leq |Z| \leq 2m$ **do**

$\quad\quad$ Play action $a^t = \pi^k(s^t)$, observe next state $s^{t+1}$ and earn reward $r^t = \frac{1}{\ell} \sum_{j=1}^\ell r_j^t$.

$\quad\quad$ Update in-episode counters and reward summation variables:

$\quad\quad$ **for** $Z \subseteq \{1, \ldots, n\}$ such that $m \leq |Z| \leq 2m$ and $i = 1, \ldots, d$ and $j = 1, \ldots, \ell$ **do**

$\quad\quad\quad$ $\nu_Z^k((s^t, a^t)[Z]) \leftarrow \nu_Z^k((s^t, a^t)[Z]) + 1$.

$\quad\quad\quad$ $\nu_{i,Z}^k((s^t, a^t)[Z], s^{t+1}[i]) \leftarrow \nu_{i,Z}^k((s^t, a^t)[Z], s^{t+1}[i]) + 1$.

$\quad\quad\quad$ $r_{j,Z}((s^t, a^t)[Z]) \leftarrow r_{j,Z}((s^t, a^t)[Z]) + r_j^t$.

$\quad\quad$ **end for**

$\quad\quad$ advance time: $t \leftarrow t + 1$.

$\quad$ **end while**

**end for**

---

---

**Algorithm 4** Eliminate Inconsistent Scopes

---

**# Eliminate Inconsistent Transition Scopes**

**for** $i = 1, \ldots, d$ and $Z \in \widetilde{\mathcal{Z}}_i^{k-1}$ **do**
    **for** $Z' \subseteq \{1, \ldots, n\}$ such that $|Z'| = m$ and $v \in X[Z \cup Z']$ and $w \in S_i$ **do**
        **if** $|\bar{P}_{i,Z \cup Z'}^k(w \mid v) - \bar{P}_{i,Z}^k(w \mid v[Z])| > 2 \cdot \epsilon_{i,Z \cup Z'}^k(w \mid v)$ **then**
           $\widetilde{\mathcal{Z}}_i^k \leftarrow \widetilde{\mathcal{Z}}_i^k \setminus \{Z\}$.
        **end if**
    **end for**
**end for**

**# Eliminate Inconsistent Reward Scopes**

**for** $j = 1, \ldots, \ell$ and $Z \in \widetilde{\mathcal{R}}_j^{k-1}$ **do**
    **for** $Z' \subseteq \{1, \ldots, n\}$ such that $|Z'| = m$ and $v \in X[Z \cup Z']$ **do**
        **if** $|\bar{r}_{j,Z \cup Z'}^k(v) - \bar{r}_{j,Z}^k(v[Z])| > 2 \cdot \epsilon_{Z \cup Z'}^k(v)$ **then**
           $\widetilde{\mathcal{R}}_j^k \leftarrow \widetilde{\mathcal{R}}_j^k \setminus \{Z\}$.
        **end if**
    **end for**
**end for**

---

---

**Algorithm 5** SLF-UCRL Compute Optimistic Policy $\pi^k$

---

Construct MDP: $\widehat{M}^k = (\widehat{S}^k, \widehat{A}^k, \widehat{P}^k, \hat{r}^k)$.

Define action space: $\widehat{A}^k = A \times S \times \widetilde{\mathcal{Z}}_1^k \times \cdots \times \widetilde{\mathcal{Z}}_d^k \times \widetilde{\mathcal{R}}_1^k \times \cdots \times \widetilde{\mathcal{R}}_\ell^k$.

Define state space: $\widehat{S}^k = S \times \{0, 1, \ldots, \log n + 1\} \times S \times \widetilde{\mathcal{Z}}_1^k \times \cdots \times \widetilde{\mathcal{Z}}_d^k \times \widetilde{\mathcal{R}}_1^k \times \cdots \times \widetilde{\mathcal{R}}_\ell^k \times \Omega^{m(d+\ell)}$,
where $\Omega = \omega^n \times \omega^{n/2} \times \cdots \times \omega^2 \times \omega$ for $\omega = (\bigcup_{i=1}^d S_i) \cup (\bigcup_{i=d+1}^n A_i)$.

Define transition function $\widehat{P}^k(\tilde{s}' \mid \tilde{s}, \tilde{a}) = \prod_{\tau=1}^{3d+\ell+1+2nm(d+\ell)} \widehat{P}_\tau^k(\tilde{s}'[\tau] \mid \tilde{s}, \tilde{a})$ as follows:

- The counter factor (factor $d + 1$) counts deterministically modulo $\log n + 2$.
- The action factors (factors $d + 2$ to $3d + \ell + 2$) take the corresponding actions played by the agent when the counter is $0$, and otherwise copy the value from the corresponding factor of the previous state.
- For $i = 1, \ldots, d$ and $e = 1, \ldots, m$, consider $\Omega_{i,e} \in \omega^n \times \omega^{n/2} \times \cdots \times \omega^2 \times \omega$ which is the $(i-1)m + e$ copy of $\Omega$. When the counter is $0$ it gets $(s, a)$, i.e., $\Omega_{i,e} = (s, a, \perp)$. When the counter is $1$, we take $(s, a)$ from $\omega^n$ and map them to $\omega^{n/2}$ while eliminating half of the factors in consideration with the consistent scope $Z_i$ chosen by the policy (stored in factor $2d + 1 + i$ of the state). This continues for $\log n$ steps until the last $\omega$ contains $(s, a)[Z_i][e]$.
- For $j = 1, \ldots, \ell$ and $e = 1, \ldots, m$, $\Omega_{j,e} \in \omega^n \times \omega^{n/2} \times \cdots \times \omega^2 \times \omega$ is the $(d+j-1)m + e$ copy of $\Omega$. It is handled similarly to the previous item, but considers the reward consistent scope $z_j$ chosen by the policy (stored in factor $3d + 1 + j$ of the state).
- For $i = 1, \ldots, d$, the $i$-th factor is taken from factor $i$ of the previous state when the counter is not $\log n + 1$, and otherwise performs the optimistic transition of factor $i$. Denote the value in the last factor of $\Omega_{i,e}$ by $v_e$, the policy's chosen scope by $Z_i$ (stored in factor $2d + 1 + i$ of the state) and the policy's chosen next state direction by $s_i'$ (stored in factor $d + 1 + i$ of the state). Then, the probability that factor $i$ transitions to $w_i \in S_i$ is

$$\bar{P}_{i,Z_i}^k(w_i \mid v_1, \ldots, v_m) - \mathcal{W}_{i,Z_i}^k(w_i \mid v_1, \ldots, v_m)$$
$$+ \mathbb{I}\{w_i = s_i'\} \cdot \sum_{w \in S_i} \mathcal{W}_{i,Z_i}^k(w \mid v_1, \ldots, v_m).$$

Define reward function $\hat{r}^k$ that is always $0$ except for the following case. When the counter is $\log n + 1$, for $j = 1, \ldots, \ell$, denote by $v_{j,e}$ the last $\omega$ in $\Omega_{j,e}$ and by $z_j$ scope chosen by the policy (stored in factor $3d + 1 + j$ of the state). Then, the $j$-th reward is: $\min\{1, \bar{r}_{j,z_j}^k(v_1, \ldots, v_m) + \epsilon_{z_j}^k(v_1, \ldots, v_m)\}$.

Compute optimal policy $\hat{\pi}^k$ of $\widehat{M}^k$ using oracle.
Extract optimistic policy: $\pi^k(s) = \hat{\pi}^k((s, 0, \perp))[1]$.

---

# B  Proof of Theorem 1

**Remark** (Unknown scope size). In this paper we assume that the learner knows a bound $m$ on the scope size in advance. However, in many applications such a bound is not available, and we are required to perform feature selection. Structure learning with unknown scope size was previously studied by Chakraborty and Stone [2011], Guo and Brunskill [2017], but as shown by the latter, it encompasses an inherent difficulty when approached without any additional assumptions. It is an interesting open problem whether additional assumptions are indeed necessary, but here we argue that under the strong assumptions made by previous works, our algorithm keeps a similar regret bound. Chakraborty and Stone [2011] assume that planning with an empirical model with insufficiently large scope size results in $\epsilon$ smaller gain than the actual one. In this case, we can keep an estimate $\tilde{m}$ of the scope size and plan twice in each episode, once with $\tilde{m}$ and once with $2\tilde{m}$. If there is a gap of more than $\epsilon$ between the gains, we double our estimate. Similarly to the doubling trick used in multi-arm bandit, this adds a constant factor (independent of $T$) to the regret. Guo and Brunskill [2017] make a similar assumption (but regarding empirical estimates of the transitions) that can be handled similarly.

**Remark** (Variable scopes sizes). For simplicity, we assume that there is a uniform bound $m$ on the scope sizes of all factors. However, our algorithm readily extends to variable scope sizes, i.e., a bound $m_i$ on the scope size of factor $i$. Without any changes to the algorithm (just setting different scope sizes for different factors), our algorithm keeps a regret bound of the same order in which the dependence in $m$ is replaced with a dependence in $m_i$ for each factor $i$.

## B.1  Bellman Equations

Define the *bias* of state $s \in S$ as follows,

$$h(M, s) = \mathbb{E}\left[\sum_{t=1}^{\infty}\big(r(s^t, \pi^\star(s^t)) - \lambda^\star(M)\big) \mid s^1 = s\right].$$

The bias vector $h(M, \cdot)$ satisfies the following Bellman optimality equations (see Puterman [1994]),

$$h(M, s) + \lambda^\star(M) = r(s, \pi^\star(s)) + \sum_{s' \in S} P(s' \mid s, \pi^\star(s))h(M, s') \quad \forall s \in S.$$

We often use the notation $h(s)$ for $h(M, s)$.

## B.2  Failure Events

We start by defining the failure events and prove that they occur with probability at most $\delta$. When the failure events do not occur, we say that we are outside the failure event.

- $F^r$ is the event that some empirical estimate of the reward function is far from its expectation. That is, there exist a time $t$, a reward factor $j$, a scope $Z$ and a value $v \in X[Z_j^r \cup Z]$ such that
$$|\bar{r}_{j,Z_j^r \cup Z}^t(v) - r_j(v[Z_j^r])| > \epsilon_{Z_j^r \cup Z}^t(v).$$
  Notice that the additional scope $Z$ has no influence because the $j$-th factor only depends on the scope $Z_j^r$. Thus, by Hoeffding inequality and a union bound the probability of $F^r$ is at most $\delta/5$.

- $F^P$ is the event that some empirical estimate of the transition function is far from its expectation. That is, there exist a time $t$, a factor $i$, a scope $Z$, a value $v \in X[Z_i^P \cup Z]$ and a value $w \in S_i$ such that
$$|\bar{P}_{i,Z_i^P \cup Z}^t(w \mid v) - P_i(w \mid v[Z_i^P])| > \epsilon_{i,Z_i^P \cup Z}^t(w \mid v).$$
  Notice that the additional scope $Z$ has no influence because the $i$-th factor only depends on the scope $Z_i^P$. Thus, by Bernstein inequality and a union bound the probability of $F^P$ is at most $\delta/5$.

- $F_{Az}^r$ is the event that
$$\sum_{t=1}^{T}\big(r(s^t, a^t) - r^t\big) > 5\sqrt{T \log \frac{10T}{\delta}}.$$
  By Azuma inequality the probability of $F_{Az}^r$ is at most $\delta/5$.

- $F_{Az}^P$ is the event that

$$\sum_{k=1}^{K}\sum_{t=t_k}^{t_{k+1}-1}\Big(\sum_{s'\in S}P(s'\mid s^t,a^t)h^k(s')-h^k(s^{t+1})\Big) > 5D\sqrt{T\log\frac{10T}{\delta}},$$

  where $h^k(s)=h(\widetilde{M}^k,s)$. By Azuma inequality the probability of $F_{Az}^P$ is at most $\delta/5$.

We define the failure event $F = F^r \cup F^P \cup F_{Az}^P \cup F_{Az}^P$, and by a union bound it occurs with probability at most $\delta$. From now on, we analyze the regret outside the failure events and therefore our regret holds with probability at least $1-\delta$.

**Remark.** Notice that outside the failure events the scopes $Z_1^P,\dots,Z_d^P$ and $Z_1^r,\dots,Z_\ell^r$ are always consistent because:

$$\begin{aligned}
\big|\bar{P}_{i,Z_i^P\cup Z}^t(w\mid v)-\bar{P}_{i,Z_i^P}^t(w\mid v[Z])\big| &\le \big|\bar{P}_{i,Z_i^P\cup Z}^t(w\mid v)-P_i(w\mid v[Z])\big|\\
&\quad+\big|P_i(w\mid v[Z])-\bar{P}_{i,Z_i^P}^t(w\mid v[Z])\big|\\
&\le \epsilon_{i,Z_i^P\cup Z}^t(w\mid v)+\epsilon_{i,Z_i^P}^t(w\mid v[Z]) \le 2\cdot\epsilon_{i,Z_i^P\cup Z}^t(w\mid v).
\end{aligned}$$

## B.3 Regret decomposition

Denote $\lambda^\star = \lambda^\star(M)$ and $\lambda^k = \lambda^\star(\widetilde{M}^k)$. Next, we decompose the total regret into the regret in each episode. Then, we further decompose it as follows:

$$\begin{aligned}
\mathrm{Reg}_T(M) &= \sum_{t=1}^{T}(\lambda^\star - r^t)\\
&= \sum_{t=1}^{T}(\lambda^\star - r(s^t,a^t)) + \sum_{t=1}^{T}(r(s^t,a^t)-r^t)\\
&\le \sum_{t=1}^{T}(\lambda^\star - r(s^t,a^t)) + O\Big(\sqrt{T\log\frac{T}{\delta}}\Big) \quad\quad (3)\\
&= \sum_{k=1}^{K}\sum_{t=t_k}^{t_{k+1}-1}(\lambda^\star - r(s^t,a^t)) + O\Big(\sqrt{T\log\frac{T}{\delta}}\Big)\\
&= \sum_{k=1}^{K}\sum_{t=t_k}^{t_{k+1}-1}(\lambda^\star - \lambda^k) \quad\quad (4)\\
&\quad + \sum_{k=1}^{K}\sum_{t=t_k}^{t_{k+1}-1}(\lambda^k - r(s^t,\pi^k(s^t))) \quad\quad (5)\\
&\quad + O\Big(\sqrt{T\log\frac{T}{\delta}}\Big),
\end{aligned}$$

where Eq. (3) holds outside the failure event (by event $F_{Az}^r$). Term (4) is the difference between the optimal gain in the actual MDP and the optimistic MDP, and is bounded by $0$ using optimism in Appendix B.4. Term (5) is the deviation of the actual sum of rewards from its expected value in the optimistic MDP, and is bounded by concentration arguments in Appendix B.5.

The theorem then follows from the combination of these two bounds, and because the true MDP $M$ is in the confidence sets of all episodes with probability at least $1-\delta$, by Appendix B.2.

## B.4 Optimism

**Lemma 4.** *For any policy $\pi : S \to A$ and any vector $h \in \mathbb{R}^{|S|}$, let $\tilde{\pi} : S \to A \times S \times \widetilde{\mathcal{Z}}_1^k \times \cdots \times \widetilde{\mathcal{Z}}_d^k \times \widetilde{\mathcal{R}}_1^k \times \cdots \times \widetilde{\mathcal{R}}_\ell^k$ be the policy satisfying $\tilde{\pi}(s) = (\pi(s), s^\star, Z_1^P,\dots,Z_d^P, Z_1^r,\dots,Z_\ell^r)$ where $s^\star = \arg\max_{s\in S} h(s)$. Then, outside the failure event,*

$$\sum_{s'\in S}\big(\widetilde{P}^k(s'\mid s,\tilde{\pi}(s))-P(s'\mid s,\pi(s))\big)h(s')\ge 0 \quad \forall s\in S.$$

*Proof.* Fix $s \in S$ and denote $x = (s, \pi(s))$. For every $i = 1, \ldots, d$ and $w \in S_i$, define $P_i^-(w \mid x[Z_i^P]) = \bar{P}_{i,Z_i^P}^k(w \mid x[Z_i^P]) - \mathcal{W}_{i,Z_i^P}^k(w \mid x[Z_i^P])$, and notice that $P^-(s' \mid x) \leq P(s' \mid x)$ outside the failure event by event $F^P$. Next, define $\alpha(s' \mid x) \overset{\text{def}}{=} \bar{P}^k(s' \mid x) - P(s' \mid x)$ and $\alpha^-(s' \mid x) \overset{\text{def}}{=} \bar{P}^k(s' \mid x) - P^-(s' \mid x)$, and notice that $\alpha(s' \mid x) \leq \alpha^-(s' \mid x)$.

Denote $H = \max_{s \in S} h(s)$. By construction of the optimistic transition function,

$$
\begin{aligned}
\sum_{s' \in S} \widetilde{P}^k(s' \mid x) h(s') &= \sum_{s' \in S} P^-(s' \mid x) h(s') + H\Big(1 - \sum_{s' \in S} P^-(s' \mid x)\Big) \\
&= \sum_{s' \in S} P^-(s' \mid x) h(s') + H \sum_{s' \in S} \alpha^-(s' \mid x) \\
&= \sum_{s' \in S} (\bar{P}^k(s' \mid x) - \alpha^-(s' \mid x)) h(s') + H \alpha^-(s' \mid x) \\
&= \sum_{s' \in S} \bar{P}^k(s' \mid x) h(s') + (H - h(s')) \alpha^-(s' \mid x) \\
&\geq \sum_{s' \in S} \bar{P}^k(s' \mid x) h(s') + (H - h(s')) \alpha(s' \mid x) \\
&= \sum_{s' \in S} (\bar{P}^k(s' \mid x) - \alpha(s' \mid x)) h(s') + H \alpha(s' \mid x) \\
&= \sum_{s' \in S} P(s' \mid x) h(s') + H \sum_{s' \in S} \alpha(s' \mid x) = \sum_{s' \in S} P(s' \mid x) h(s').
\end{aligned}
$$

$\square$

**Corollary 5.** *Let* $\tilde{\pi}^\star : S \to A \times S \times \widetilde{\mathcal{Z}}_1^k \times \cdots \times \widetilde{\mathcal{Z}}_d^k \times \widetilde{\mathcal{R}}_1^k \times \cdots \times \widetilde{\mathcal{R}}_\ell^k$ *be the policy that satisfies* $\tilde{\pi}^\star(s) = (\pi^\star(s), s^\star, Z_1^P, \ldots, Z_d^P, Z_1^r, \ldots, Z_\ell^r)$, *where* $s^\star = \max_{s \in S} h(M, s)$. *Then, outside the failure event,* $\lambda(\widetilde{M}^k, \tilde{\pi}^\star, s_1) \geq \lambda^\star$ *for any starting state* $s_1$.

*Proof.* Let $\rho(\cdot) \in \mathbb{R}^{|S|}$ be the vector of stationary distribution for playing policy $\tilde{\pi}^\star$ in $\widetilde{M}^k$. By definition of the average reward we have,

$$
\begin{aligned}
\lambda(\widetilde{M}^k, \tilde{\pi}^\star, s_1) - \lambda^\star &= \sum_{s \in S} \rho(s) \tilde{r}^k(s, \tilde{\pi}^\star(s)) - \lambda^\star \\
&= \sum_{s \in S} \rho(s) \big(\tilde{r}^k(s, \tilde{\pi}^\star(s)) - \lambda^\star\big) \\
&\geq \sum_{s \in S} \rho(s) \big(r(s, \pi^\star(s)) - \lambda^\star\big) \\
&= \sum_{s \in S} \rho(s) \Big(h(M, s) - \sum_{s' \in S} P(s' \mid s, \pi^\star(s)) h(M, s')\Big) \\
&= \sum_{s \in S} \rho(s) \Big(\sum_{s' \in S} \widetilde{P}^k(s' \mid s, \tilde{\pi}^\star(s)) - \sum_{s' \in S} P(s' \mid s, \pi^\star(s))\Big) h(M, s') \geq 0,
\end{aligned}
$$

where the first inequality is by definition of the reward function in $\widetilde{M}^k$ and event $F^r$, and the following equality is by the Bellman equations. The last equality follows because $\rho$ is the stationary distribution of $\tilde{\pi}^\star$ is $\widetilde{M}^k$ and therefore $\rho(s') = \sum_{s \in S} \rho(s) \widetilde{P}^k(s' \mid s, \tilde{\pi}^\star(s))$. The final inequality is by Lemma 4. $\square$

### B.5 Bounding the Deviation

Denote by $\nu^k(s, a)$ the number of visits to state-action pair $(s, a)$ in episode $k$, and let $\nu^k(s) = \nu^k(s, \pi^k(s))$ and

$$
\Delta_k = \sum_{s \in S} \sum_{a \in A} \nu^k(s, a)(\lambda^k - r(s, a)) = \sum_{s \in S} \nu^k(s)(\lambda^k - r(s, \pi^k(s))).
$$

Thus: $(5) = \sum_{k=1}^{K}\sum_{t=t_k}^{t_{k+1}-1}(\lambda^k - r(s^t, \pi^k(s^t))) = \sum_{k=1}^{K}\Delta_k.$

We now focus on a single episode $k$. By the Bellman equations in the optimistic model $\widetilde{M}^k$ we have,

$$
\begin{aligned}
\Delta_k &= \sum_{s\in S}\nu^k(s)(\lambda^k - r(s, \pi^k(s)))\\
&= \sum_{s\in S}\nu^k(s)(\lambda^k - \tilde{r}^k(s, \tilde{\pi}^k(s))) + \sum_{s\in S}\nu^k(s)(\tilde{r}^k(s, \tilde{\pi}^k(s)) - r(s, \pi^k(s)))\\
&= \sum_{s\in S}\nu^k(s)\Big(\sum_{s'\in S}\widetilde{P}^k(s'\mid s, \tilde{\pi}^k(s))h^k(s') - h^k(s)\Big) + \sum_{s\in S}\nu^k(s)(\tilde{r}^k(s, \tilde{\pi}^k(s)) - r(s, \pi^k(s)))\\
&= \sum_{s\in S}\nu^k(s)\sum_{s'\in S}h^k(s')\big(\widetilde{P}^k(s'\mid s, \tilde{\pi}^k(s)) - P(s'\mid s, \pi^k(s))\big)\\
&\quad + \sum_{s\in S}\nu^k(s)\Big(\sum_{s'\in S}P(s'\mid s, \pi^k(s))h^k(s') - h^k(s)\Big) + \sum_{s\in S}\nu^k(s)(\tilde{r}^k(s, \tilde{\pi}^k(s)) - r(s, \pi^k(s)))\\
&\leq D\sum_{s\in S}\nu^k(s)\|\widetilde{P}^k(\cdot\mid s, \tilde{\pi}^k(s)) - P(\cdot\mid s, \pi^k(s))\|_1 & (6)\\
&\quad + \sum_{t=t_k}^{t_{k+1}-1}\Big(\sum_{s'\in S}P(s'\mid s^t, a^t)h^k(s') - h^k(s^t)\Big) & (7)\\
&\quad + \sum_{s\in S}\nu^k(s)(\tilde{r}^k(s, \tilde{\pi}^k(s)) - r(s, \pi^k(s))), & (8)
\end{aligned}
$$

where $h^k(s) = h(\widetilde{M}^k, s)$, and the last inequality follows from standard arguments [Jaksch et al., 2010] since $h^k(s) \leq D$ similarly to Lemma 3 in Xu and Tewari [2020]. We now bound each term separately.

**Term (7).** We can add and subtract $h^k(s^{t+1})$ to term (7), and then when we sum it across all episodes, we obtain a telescopic sum that is bounded by $KD$ for all episode switches, plus a martingale difference sequence bounded by event $F_{Az}^P$. That is,

$$
\sum_{k=1}^{K}\sum_{t=t_k}^{t_{k+1}-1}\Big(\sum_{s'\in S}P(s'\mid s^t, a^t)h^k(s') - h^k(s^t)\Big) \leq O\Big(D\sqrt{T\log\frac{T}{\delta}} + KD\Big).
$$

**Term (6).** Let $\lesssim$ represent $\leq$ up to numerical constants, and denote $x = (s, \pi^k(s))$, $\tilde{x} = (s, \tilde{\pi}^k(s))$ and $\tilde{\pi}^k(s) = (\pi^k(s), s_n^k(s), Z_1^k(s), \ldots, Z_d^k(s), z_1^k(s), \ldots, z_\ell^k(s))$. We can bound the distance between $P$ and $\widetilde{P}^k$ by the sum of distances between $P_i$ and $\widetilde{P}_i^k$ [Osband and Van Roy, 2014], i.e.,

$$
\begin{aligned}
\|\widetilde{P}^k(\cdot\mid\tilde{x}) - P(\cdot\mid x)\|_1 &\leq \sum_{i=1}^{d}\big\|\widetilde{P}_i^k\big(\cdot\mid x[Z_i^k(s)]\big) - P_i\big(\cdot\mid x[Z_i^P]\big)\big\|_1\\
&\leq \sum_{i=1}^{d}\big\|\widetilde{P}_i^k\big(\cdot\mid x[Z_i^k(s)]\big) - \bar{P}_{i,Z_i^k(s)}^k\big(\cdot\mid x[Z_i^k(s)]\big)\big\|_1 & (9)\\
&\quad + \sum_{i=1}^{d}\big\|\bar{P}_{i,Z_i^k(s)}^k\big(\cdot\mid x[Z_i^k(s)]\big) - \bar{P}_{i,Z_i^P}^k\big(\cdot\mid x[Z_i^P]\big)\big\|_1 & (10)\\
&\quad + \sum_{i=1}^{d}\big\|\bar{P}_{i,Z_i^P}^k\big(\cdot\mid x[Z_i^P]\big) - P_i\big(\cdot\mid x[Z_i^P]\big)\big\|_1 & (11)\\
&\leq \sum_{i=1}^{d}\sum_{w\in S_i}\epsilon_{i,Z_i^k(s)}^k(w\mid x[Z_i^k(s)]) + 4\cdot\epsilon_{i,Z_i^P\cup Z_i^k(s)}^k(w\mid x[Z_i^P\cup Z_i^k(s)]) + \epsilon_{i,Z_i^P}^k(w\mid x[Z_i^P])\\
&\lesssim \sum_{i=1}^{d}\sqrt{\frac{|S_i|\log\big(\frac{dLWT}{\delta}\big)}{\max\{N_{Z_i^P\cup Z_i^k(s)}^k(x[Z_i^P\cup Z_i^k(s)]), 1\}}} + \frac{|S_i|\log\big(\frac{dLWT}{\delta}\big)}{\max\{N_{Z_i^P\cup Z_i^k(s)}^k(x[Z_i^P\cup Z_i^k(s)]), 1\}},
\end{aligned}
$$

where term (9) is bounded by the construction of the optimistic MDP, and term (11) is bounded by event $F^P$. Term (10) is bounded because the policy $\tilde{\pi}^k$ chooses only consistent scopes. Since $Z_i^k(s)$ and $Z_i^P$ are both consistent (outside the failure event), we have that $\bar{P}^k_{i,Z_i^k(s)}$ and $\bar{P}^k_{i,Z_i^P}$ are both close to $\bar{P}^k_{i,Z_i^P \cup Z_i^k(s)}$. Thus, we can bound term (6) as follows

$$
\begin{aligned}
\sum_{k=1}^{K} (6) &\leq D \sum_{k=1}^{K} \sum_{s \in S} \nu^k(s) \|\widetilde{P}^k(\cdot \mid s, \tilde{\pi}^k(s)) - P(\cdot \mid s, \pi^k(s))\|_1 \\
&\lesssim D \sum_{k=1}^{K} \sum_{s \in S} \sum_{i=1}^{d} \nu^k(s) \sqrt{\frac{|S_i| \log\left(\frac{dLWT}{\delta}\right)}{\max\{N^k_{Z_i^P \cup Z_i^k(s)}(x[Z_i^P \cup Z_i^k(s)]), 1\}}} \\
&\quad + D \sum_{k=1}^{K} \sum_{s \in S} \sum_{i=1}^{d} \frac{\nu^k(s)|S_i| \log\left(\frac{dLWT}{\delta}\right)}{\max\{N^k_{Z_i^P \cup Z_i^k(s)}(x[Z_i^P \cup Z_i^k(s)]), 1\}} \\
&\lesssim D \sum_{k=1}^{K} \sum_{i=1}^{d} \sum_{Z:|Z|=m} \sum_{v \in X[Z_i^P \cup Z]} \nu^k_{Z_i^P \cup Z}(v) \sqrt{\frac{|S_i| \log\left(\frac{dLWT}{\delta}\right)}{\max\{N^k_{Z_i^P \cup Z}(v), 1\}}} \\
&\quad + D \sum_{k=1}^{K} \sum_{i=1}^{d} \sum_{Z:|Z|=m} \sum_{v \in X[Z_i^P \cup Z]} \frac{\nu^k_{Z_i^P \cup Z}(v)|S_i| \log\left(\frac{dLWT}{\delta}\right)}{\max\{N^k_{Z_i^P \cup Z}(v), 1\}} \\
&\lesssim D \sum_{i=1}^{d} \sum_{Z:|Z|=m} \sum_{v \in X[Z_i^P \cup Z]} \sqrt{N^{K+1}_{Z_i^P \cup Z}(v)|S_i| \log\left(\frac{dLWT}{\delta}\right)} + |S_i| \log\left(\frac{dLWT}{\delta}\right) \log T \\
&\lesssim D \sum_{i=1}^{d} \sum_{Z:|Z|=m} \sqrt{|X[Z_i^P \cup Z]| \sum_{v \in X[Z_i^P \cup Z]} N^{K+1}_{Z_i^P \cup Z}(v)|S_i| \log\left(\frac{dLWT}{\delta}\right)} \\
&\quad + D \sum_{i=1}^{d} \sum_{Z:|Z|=m} \sum_{v \in X[Z_i^P \cup Z]} |S_i| \log\left(\frac{dLWT}{\delta}\right) \log T \\
&\lesssim D \sum_{i=1}^{d} \sum_{Z:|Z|=m} \sqrt{|X[Z_i^P \cup Z]||S_i|T \log\left(\frac{dLWT}{\delta}\right)} \\
&\quad + D \sum_{i=1}^{d} \sum_{Z:|Z|=m} |X[Z_i^P \cup Z]||S_i| \log\left(\frac{dLWT}{\delta}\right) \log T,
\end{aligned}
$$

where the third inequality follows from our construction of the episodes as doubling number of visits to some scope-sized state-action pair (specifically, from Lemma 19 in Jaksch et al. [2010] and Lemma B.18 in Rosenberg et al. [2020]), the forth inequality follows from Jensen's inequality, and the last one because $\sum_{v \in X[Z_i^P \cup Z]} N^{K+1}_{Z_i^P \cup Z}(v) \leq T$.

**Term (8).** We can bound the distance between $r$ and $\widetilde{r}^k$ by the sum of distances between $r_j$ and $\widetilde{r}^k_j$,

$$\widetilde{r}^k(s, \widetilde{\pi}^k(s)) - r(s, \pi^k(s)) = \frac{1}{\ell} \sum_{j=1}^{\ell} \widetilde{r}^k_j(\widetilde{x}[z^k_j(s)]) - r_j(x[Z^r_j])$$

$$= \underbrace{\frac{1}{\ell} \sum_{j=1}^{\ell} \widetilde{r}^k_j(\widetilde{x}[z^k_j(s)]) - \bar{r}_j(x[z^k_j(s)])}_{(a)} + \underbrace{\frac{1}{\ell} \sum_{j=1}^{\ell} \bar{r}^k_j(x[z^k_j(s)]) - \bar{r}_j(x[Z^r_j])}_{(b)}$$

$$+ \underbrace{\frac{1}{\ell} \sum_{j=1}^{\ell} \bar{r}^k_j(x[Z^r_j]) - r_j(x[Z^r_j])}_{(c)}$$

$$\leq \frac{1}{\ell} \sum_{j=1}^{\ell} \epsilon^k_{z^k_j(s)}(x[z^k_j(s)]) + 4 \cdot \epsilon^k_{Z^r_j \cup z^k_j(s)}(x[Z^r_j \cup z^k_j(s)]) + \epsilon^k_{Z^r_j}(x[Z^r_j])$$

$$\lesssim \frac{1}{\ell} \sum_{j=1}^{\ell} \sqrt{\frac{\log\left(\frac{dLWT}{\delta}\right)}{\max\{N^k_{Z^r_j \cup z^k_j(s)}(x[Z^r_j \cup z^k_j(s)]), 1\}}},$$

where (a) is bounded by the construction of the optimistic MDP, and (c) is bounded by event $F^r$. (b) is bounded because the policy $\widetilde{\pi}^k$ chooses only consistent reward scopes. Since $z^k_j(s)$ and $Z^r_j$ are both consistent (outside the failure event), we have that $\bar{r}^k_{j,z^k_j(s)}$ and $\bar{r}^k_{j,Z^r_j}$ are both close to $\bar{r}^k_{j,Z^r_j \cup z^k_j(s)}$. Thus, we can bound term (8) as follows

$$\sum_{k=1}^{K} (8) = \frac{1}{\ell} \sum_{k=1}^{K} \sum_{s \in S} \sum_{j=1}^{\ell} \nu^k(s)(\widetilde{r}^k(s, \widetilde{\pi}^k(s)) - r(s, \pi^k(s)))$$

$$\lesssim \frac{1}{\ell} \sum_{k=1}^{K} \sum_{s \in S} \sum_{j=1}^{\ell} \nu^k(s) \sqrt{\frac{\log\left(\frac{dLWT}{\delta}\right)}{\max\{N^k_{Z^r_j \cup z^k_j(s)}(x[Z^r_j \cup z^k_j(s)]), 1\}}}$$

$$\lesssim \frac{1}{\ell} \sum_{k=1}^{K} \sum_{j=1}^{\ell} \sum_{Z:|Z|=m} \sum_{v \in X[Z^r_j \cup Z]} \nu^k_{Z^r_j \cup Z}(v) \sqrt{\frac{\log\left(\frac{dLWT}{\delta}\right)}{\max\{N^k_{Z^r_j \cup Z}(v), 1\}}}$$

$$\lesssim \frac{1}{\ell} \sum_{j=1}^{\ell} \sum_{Z:|Z|=m} \sum_{v \in X[Z^r_j \cup Z]} \sqrt{N^{K+1}_{Z^r_j \cup Z}(v) \log\left(\frac{dLWT}{\delta}\right)}$$

$$\lesssim \frac{1}{\ell} \sum_{j=1}^{\ell} \sum_{Z:|Z|=m} \sqrt{|X[Z^r_j \cup Z]| \sum_{v \in X[Z^r_j \cup Z]} N^{K+1}_{Z^r_j \cup Z}(v) \log\left(\frac{dLWT}{\delta}\right)}$$

$$\lesssim \frac{1}{\ell} \sum_{j=1}^{\ell} \sum_{Z:|Z|=m} \sqrt{|X[Z^r_j \cup Z]| T \log\left(\frac{dLWT}{\delta}\right)},$$

where the third inequality follows from our construction of the episodes as doubling number of visits to some scope-sized state-action pair (specifically, from Lemma 19 in Jaksch et al. [2010] and Lemma B.18 in Rosenberg et al. [2020]), the forth inequality follows from Jensen's inequality, and the last one because $\sum_{v \in X[Z^r_j \cup Z]} N^{K+1}_{Z^r_j \cup Z}(v) \leq T$.

## B.6 Putting Everything Together

Taking the bounds on all the terms, and noting that the failure event occurs with probability at most $\delta$, gives the following regret bound.

$$
\begin{aligned}
\mathrm{Reg}_T(M) &\lesssim \sqrt{T\log\frac{T}{\delta}} + D\sqrt{T\log\frac{T}{\delta}} + KD + \frac{1}{\ell}\sum_{j=1}^{\ell}\sum_{Z:|Z|=m}\sqrt{|X[Z_j^r \cup Z]|T\log\Big(\frac{dLWT}{\delta}\Big)} \\
&\quad + D\sum_{i=1}^{d}\sum_{Z:|Z|=m}\sqrt{|X[Z_i^P \cup Z]||S_i|T\log\Big(\frac{dLWT}{\delta}\Big)} \\
&\quad + D\sum_{i=1}^{d}\sum_{Z:|Z|=m}\sum_{v\in X[Z_i^P\cup Z]}|S_i|\log\Big(\frac{dLWT}{\delta}\Big)\log T \\
&\lesssim \sum_{i=1}^{d}\sum_{Z:|Z|=m}D\sqrt{|X[Z_i^P \cup Z]||S_i|T\log\Big(\frac{dLWT}{\delta}\Big)} \\
&\quad + \frac{1}{\ell}\sum_{j=1}^{\ell}\sum_{Z:|Z|=m}\sqrt{|X[Z_j^r \cup Z]|T\log\Big(\frac{dLWT}{\delta}\Big)} \\
&\quad + \sum_{i=1}^{d}\sum_{Z:|Z|=m}D|X[Z_i^P \cup Z]||S_i|\log^2\Big(\frac{dLWT}{\delta}\Big) \\
&\quad + \sum_{Z:|Z|=m}\sum_{Z':|Z'|=m}D|X[Z\cup Z']|\log T \\
&\lesssim \binom{n}{m}dD\sqrt{L^2WT\log\Big(\frac{dLWT}{\delta}\Big)} + \binom{n}{m}dDL^2W\log^2\Big(\frac{dLWT}{\delta}\Big) \\
&\quad + \binom{n}{m}^2 DL^2\log T,
\end{aligned}
$$

where the second inequality follows because there are at most $\log T$ episodes for each pair of scopes $Z \neq Z'$ of size $m$ and $v \in X[Z \cup Z']$.

## C  The NFA-DORL Algorithm

---

**Algorithm 6** NFA-DORL

---

**Input:** confidence parameter $\delta$, scopes $\{Z_i^P\}_{i=1}^d$, reward scopes $\{Z_j^r\}_{j=1}^\ell$, state space $S = \{S_i\}_{i=1}^d$, action space $A$.

**# Initialization**

Initialize total visit counters $N$, in-episode visit counters $\nu$ and reward summation variables $r$:

**for** $a \in A$ and $j = 1, \ldots, \ell$ and $v_j \in S[Z_j^r]$ and $i = 1, \ldots, d$ and $v_i \in S[Z_i^P]$ and $w \in S_i$ **do**

$\quad r_{j,Z_j^r}(v_j,a) \leftarrow 0, N_{Z_j^r}^0(v_j,a) \leftarrow 0, \nu_{Z_j^r}^0(v_j,a) \leftarrow 0, N_{i,Z_i^P}^0(v_i,a,w) \leftarrow 0, \nu_{i,Z_i^P}^0(v_i,a,w) \leftarrow 0, N_{Z_i^P}^0(v_i,a) \leftarrow 0, \nu_{Z_i^P}^0(v_i,a) \leftarrow 0.$

**end for**

Initialize time steps counter: $t \leftarrow 1$, and observe initial state $s^1$.

**for** $k = 1, 2, \ldots$ **do**

$\quad$ **# Start New Episode**

$\quad$ Set episode starting time: $t_k \leftarrow t$.

$\quad$ **for** $a \in A$ and $j = 1, \ldots, \ell$ and $v_j \in S[Z_j^r]$ and $i = 1, \ldots, d$ and $v_i \in S[Z_i^P]$ and $w \in S_i$ **do**

$\quad\quad$ Update visit counters: $\nu_{Z_i^P}^k(v_i,a) \leftarrow 0, \nu_{Z_j^r}^k(v_j,a) \leftarrow 0, \nu_{i,Z_i^P}^k(v_i,a,w) \leftarrow 0, N_{Z_i^P}^k(v_i,a) \leftarrow N_{Z_i^P}^{k-1}(v_i,a) + \nu_{Z_i^P}^{k-1}(v_i,a), N_{Z_j^r}^k(v_j,a) \leftarrow N_{Z_j^r}^{k-1}(v_j,a) + \nu_{Z_j^r}^{k-1}(v_j,a), N_{i,Z_i^P}^k(v_i,a,w) \leftarrow N_{i,Z_i^P}^{k-1}(v_i,a,w) + \nu_{i,Z_i^P}^{k-1}(v_i,a,w).$

$\quad\quad$ Compute empirical transitions and rewards:

$$\bar{P}_{i,Z_i^P}^k(w \mid v_i,a) = \frac{N_{i,Z_i^P}^k(v_i,a,w)}{\max\{N_{Z_i^P}^k(v_i,a),1\}} \quad ; \quad \bar{r}_{j,Z_j^r}^k(v_j,a) = \frac{r_{j,Z_j^r}(v_j,a)}{\max\{N_{Z_j^r}^k(v_j,a),1\}}.$$

$\quad\quad$ Set confidence bounds ($\tau^k = \log \frac{6dWLt_k}{\delta}$):

$$\epsilon_{i,Z_i^P}^k(w \mid v_i,a) = \sqrt{\frac{18\bar{P}_{i,Z_i^P}^k(w \mid v_i,a)\tau^k}{\max\{N_{Z_i^P}^k(v_i,a),1\}}} + \frac{18\tau^k}{\max\{N_{Z_i^P}^k(v_i,a),1\}}$$

$$\epsilon_{Z_j^r}^k(v_j,a) = \sqrt{\frac{18\tau^k}{\max\{N_{Z_j^r}^k(v_j,a),1\}}}$$

$$\mathcal{W}_{i,Z_i^P}^k(w \mid v_i,a) = \min\{\epsilon_{i,Z_i^P}^k(w \mid v_i,a), \bar{P}_{i,Z_i^P}^k(w \mid v_i,a)\}.$$

$\quad$ **end for**

$\quad$ Construct optimistic MDP $\widetilde{M}^k$ and compute optimistic policy $\pi^k$ (Algorithm 7).

$\quad$ **# Execute Policy**

$\quad$ **while** $\nu_Z^k(s^t[Z], \pi^k(s^t)) < N_Z^k(s^t[Z], \pi^k(s^t))$ for every $Z \in \{Z_1^P, \ldots, Z_d^P, Z_1^r, \ldots, Z_\ell^r\}$ **do**

$\quad\quad$ Play action $a^t = \pi^k(s^t)$, observe next state $s^{t+1}$ and earn reward $r^t = \frac{1}{\ell}\sum_{j=1}^\ell r_j^t$.

$\quad\quad$ Update in-episode counters and reward summation variables:

$\quad\quad$ **for** $i = 1, \ldots, d$ and $j = 1, \ldots, \ell$ **do**

$\quad\quad\quad \nu_{Z_i^P}^k(s^t[Z_i^P],a^t) \leftarrow \nu_{Z_i^P}^k(s^t[Z_i^P],a^t) + 1, \nu_{Z_j^r}^k(s^t[Z_i^P],a^t) \leftarrow \nu_{Z_j^r}^k(s^t[Z_j^r],a^t) + 1.$

$\quad\quad\quad \nu_{i,Z_i^P}^k(s^t[Z_i^P],a^t,s^{t+1}[i]) \leftarrow \nu_{i,Z_i^P}^k(s^t[Z_i^P],a^t,s^{t+1}[i]) + 1.$

$\quad\quad\quad r_{j,Z}(s^t[Z_j^r],a^t) \leftarrow r_{j,Z_j^r}(s^t[Z_j^r],a^t) + r_j^t.$

$\quad\quad$ **end for**

$\quad\quad$ advance time: $t \leftarrow t + 1.$

$\quad$ **end while**

**end for**

---

**Algorithm 7** NFA-DORL Compute Optimistic Policy $\pi^k$

---

Construct MDP: $\widetilde{M}^k = (\widetilde{S}, \widetilde{A}, \widetilde{P}^k, \tilde{r}^k)$.

Define action space: $\widetilde{A} = A \cup (\bigcup_{i=1}^d S_i)$.

Define state space: $\widetilde{S} = S \times \{0, 1, \ldots, d+1\} \times A \times S \times \{0, 1\}$.

Define reward function for $j = 1, \ldots, \ell$:

$$\tilde{r}_j^k\big((s, h, a', s', b), a\big) = \begin{cases} \min\big\{1, \bar{r}_{j,Z_j^r}^k(s[Z_j^r], a) + \epsilon_{Z_j^r}^k(s[Z_j^r], a)\big\}, & b = 1, h = 0, a \in A \\ 0, & otherwise \end{cases}$$

Define transition function $\widetilde{P}^k\big(\tilde{s}' \mid \tilde{s}, \tilde{a}\big) = \prod_{\tau=1}^{2d+3} \widetilde{P}_\tau^k\big(\tilde{s}'[\tau] \mid \tilde{s}, \tilde{a}\big)$ as follows:

- The counter factor (factor $d+1$) counts deterministically modulo $d+2$.
- The action factor (factor $d+2$) takes the action played by the agent when the counter is 0, and otherwise copies the value from the $(d+2)$-th factor of the previous state.
- The last factor checks that all actions are legal. It starts at 1 and changes to 0 if the taken action $a$ satisfies (1) $a \notin A$ when the counter is 0 ; (2) $a \notin S_i$ when the counter is $i$ (for $i = 1, \ldots, d$).
- For $i = 1, \ldots, d$, the $i$-th factor is taken from factor $i + 1 + d$ of the previous state when the counter is $d + 1$, and otherwise copies the value from the $i$-th factor of the previous state.
- For $i = 1, \ldots, d$, the $(i+2+d)$-th factor is taken from factor $i+2+d$ of the previous state when the counter is not $i$, and otherwise performs the optimistic transition of factor $i$ (if the action is not in $S_i$ transition arbitrarily), i.e.,

$$\widetilde{P}_{i+2+d}^k\big(w_i \mid (s, i, a, s', b), w\big) = \bar{P}_{i,Z_i^P}^k(w_i \mid s[Z_i^P], a) - \mathcal{W}_{i,Z_i^P}^k(w_i \mid s[Z_i^P], a)$$
$$+ \mathbb{I}\{w_i = w\} \cdot \sum_{w' \in S_i} \mathcal{W}_{i,Z_i^P}^k(w' \mid s[Z_i^P], a).$$

Compute optimal policy $\tilde{\pi}^k$ of $\widetilde{M}^k$ using oracle.

Extract optimistic policy: $\pi^k(s) = \tilde{\pi}^k((s, 0, \perp))$.

---

# D Proof of Theorem 2

The proof relies on the MDP $M' = (\widetilde{S}, A, P', r')$ (described in Section 5) that models $M$ but stretches each time step to $d + 2$ steps. Given a trajectory $(s^t, a^t)_{t=1,\ldots,T}$ in $M$, we map it to a trajectory $(s^{t,h}, a^{t,h})_{t=1,\ldots,T,h=0,1,\ldots,d+1}$ in $M'$ as follows:

- $s^{t,0} = (s^t, 0, \bot)$ and $a^{t,0} = a^t$.

- $s^{t,1} = (s^t, 1, a^t, \bot)$ and $a^{t,1}$ is arbitrary.

- $s^{t,i+1} = (s^t, i+1, a^t, s^{t+1}[1], \ldots, s^{t+1}[i], \bot)$ for $i = 1, \ldots, d$ and $a^{t,i+1}$ is arbitrary.

Moreover, we slightly abuse notation as follows. For a policy $\pi$ in $M$, we use the same notation $\pi$ also for the policy in $M'$ that plays according to $\pi$. That is, $\pi(s^{t,0}) = \pi((s^t, 0, \bot)) = \pi(s^t)$ and $\pi(s^{t,h})$ is arbitrary for $h > 0$ as the policy has no effect in these steps.

The failure events for the algorithm are similar to Appendix B.2. Recall that $\lambda^\star(M') = \frac{\lambda^\star(M)}{d+2}$ and therefore we can write:

$$
\begin{aligned}
\mathrm{Reg}_T(M) &= \sum_{t=1}^{T} \left( \lambda^\star(M) - r^t \right) \\
&= \sum_{t=1}^{T} \left( \lambda^\star(M) - r(s^t, a^t) \right) + \sum_{t=1}^{T} \left( r(s^t, a^t) - r^t \right) \\
&\leq \sum_{t=1}^{T} \left( \lambda^\star(M) - r(s^t, a^t) \right) + O\left( \sqrt{T \log \frac{T}{\delta}} \right) \\
&= \sum_{t=1}^{T} \left( \frac{\lambda^\star(M)}{d+2} - r(s^t, a^t) \right) + \sum_{t=1}^{T} \sum_{h=1}^{d+1} \left( \frac{\lambda^\star(M)}{d+2} - 0 \right) + O\left( \sqrt{T \log \frac{T}{\delta}} \right) \\
&= \sum_{t=1}^{T} \sum_{h=0}^{d+1} \left( \lambda^\star(M') - r'(s^{t,h}, a^{t,h}) \right) + O\left( \sqrt{T \log \frac{T}{\delta}} \right) \\
&= \sum_{k=1}^{K} \sum_{t=t_k}^{t_{k+1}-1} \sum_{h=0}^{d+1} \left( \lambda^\star(M') - r'(s^{t,h}, \pi^k(s^{t,h})) \right) + O\left( \sqrt{T \log \frac{T}{\delta}} \right) \\
&\leq \sum_{k=1}^{K} \sum_{t=t_k}^{t_{k+1}-1} \sum_{h=0}^{d+1} \left( \lambda^\star(\widetilde{M}^k) - r'(s^{t,h}, \pi^k(s^{t,h})) \right) + O\left( \sqrt{T \log \frac{T}{\delta}} \right) \\
&= \sum_{k=1}^{K} \sum_{t=t_k}^{t_{k+1}-1} \sum_{h=0}^{d+1} \left( \lambda^\star(\widetilde{M}^k) - \widetilde{r}^k(s^{t,h}, \pi^k(s^{t,h})) \right) \quad (12) \\
&\quad + \sum_{k=1}^{K} \sum_{t=t_k}^{t_{k+1}-1} \sum_{h=0}^{d+1} \left( \widetilde{r}^k(s^{t,h}, \pi^k(s^{t,h})) - r'(s^{t,h}, \pi^k(s^{t,h})) \right) \quad (13) \\
&\quad + O\left( \sqrt{T \log \frac{T}{\delta}} \right),
\end{aligned}
$$

where the last inequality is by optimism which is proven similarly to Appendix B.4.

**Term (13).** Notice that the reward is zero when the counter is not 0 and therefore

$$(13) = \sum_{k=1}^{K} \sum_{t=t_k}^{t_{k+1}-1} \left( \tilde{r}^k(s^{t,0}, \pi^k(s^{t,0})) - r'(s^{t,0}, \pi^k(s^{t,0})) \right)$$

$$\leq \frac{1}{\ell} \sum_{k=1}^{K} \sum_{s \in S} \sum_{j=1}^{\ell} \nu^k(s) \left( \tilde{r}_{j,Z_j^r}^k(s[Z_j^r], \pi^k(s)) - r_j(s[Z_j^r], \pi^k(s)) + \epsilon_{Z_j^r}^k(s[Z_j^r], \pi^k(s)) \right)$$

$$\leq \frac{1}{\ell} \sum_{k=1}^{K} \sum_{s \in S} \sum_{j=1}^{\ell} \nu^k(s) \cdot 2\epsilon_{Z_j^r}^k(s[Z_j^r], \pi^k(s))$$

$$\lesssim \frac{1}{\ell} \sum_{k=1}^{K} \sum_{j=1}^{\ell} \sum_{v \in S[Z_j^r]} \sum_{a \in A} \nu_{Z_j^r}^k(v, a) \sqrt{\frac{\log \frac{dWLT}{\delta}}{\max\{N_{Z_j^r}^k(v, a), 1\}}}$$

$$\lesssim \frac{1}{\ell} \sum_{j=1}^{\ell} \sqrt{|S[Z_j^r]||A|T \log \frac{dWLT}{\delta}}.$$

**Term (12).** By the Bellman equations in the optimistic model $\widetilde{M}^k$, we can write term (12) as follows

$$(12) = \sum_{k=1}^{K} \sum_{t=t_k}^{t_{k+1}-1} \sum_{h=0}^{d+1} \left( \sum_{s' \in \widetilde{S}} \widetilde{P}^k(s' \mid s^{t,h}, \pi^k(s^{t,h})) h^k(s') - h^k(s^{t,h}) \right)$$

$$= \sum_{k=1}^{K} \sum_{t=t_k}^{t_{k+1}-1} \sum_{h=0}^{d+1} \sum_{s' \in \widetilde{S}} \left( \widetilde{P}^k(s' \mid s^{t,h}, \pi^k(s^{t,h})) - P'(s' \mid s^{t,h}, \pi^k(s^{t,h})) \right) h^k(s')$$

$$+ \sum_{t=t_k}^{t_{k+1}-1} \sum_{h=0}^{d+1} \left( \sum_{s' \in \widetilde{S}} P'(s' \mid s^{t,h}, \pi^k(s^{t,h})) h^k(s') - h^k(s^{t,h}) \right)$$

$$\lesssim D \sum_{k=1}^{K} \sum_{s \in S} \sum_{i=1}^{d} \sum_{w \in S_i} \nu^k(s) \epsilon_{i,Z_i^P}^k(s[Z_i^P], \pi^k(s), w)$$

$$+ \sum_{t=t_k}^{t_{k+1}-1} \sum_{h=0}^{d+1} \left( \sum_{s' \in \widetilde{S}} P'(s' \mid s^{t,h}, \pi^k(s^{t,h})) h^k(s') - h^k(s^{t,h}) \right)$$

$$\lesssim D \sum_{k=1}^{K} \sum_{i=1}^{d} \sum_{v \in S[Z_i^P]} \sum_{a \in A} \nu_{Z_i^P}^k(v, a) \left( \sqrt{\frac{|S_i| \log \frac{dWLT}{\delta}}{\max\{N_{Z_i^P}^k(v, a), 1\}}} + \frac{|S_i| \log \frac{dWLT}{\delta}}{\max\{N_{Z_i^P}^k(v, a), 1\}} \right)$$

$$+ KD + D\sqrt{dT \log \frac{dT}{\delta}}$$

$$\lesssim \sum_{i=1}^{d} D\sqrt{|S_i||S[Z_i^P]||A|T \log \frac{dWLT}{\delta}} + \sum_{i=1}^{d} D|S_i||S[Z_i^P]||A| \log^2 \frac{dWLT}{\delta}.$$

The first inequality follows by the definition of $P'$ and $\widetilde{P}^k$ and their factored structure. The second inequality is similar to Appendix B.5, while noting that the bias function in $\widetilde{M}^k$ is bounded by $D$. The reason is that diameter of $\widetilde{M}^k$ is $D(d+2)$, and that the bias function is always bounded by the diameter times the optimal gain (see Bartlett and Tewari [2009]).

# E   Factored MDPs with Non-Factored Actions and Unkown Structure

We now adjust our SLF-UCRL algorithm to cope with non-factored actions. The idea is similar to Section 5 – instead of choosing a factored action that contains the actual action and the optimistic choices for all the consistent scopes, this time step will be stretched across $2 + d(m + 1)$ steps in which the policy makes its choice sequentially. In the first step the policy picks the action, in steps $i(m + 1) - m$ to $i(m + 1) - 1$ it picks a consistent scope for factor $i$, step $i(m + 1)$ performs the optimistic transition of the $i$-th factor, and the last step completes the transition.

Thus, the action space of the optimistic MDP $\widetilde{M}^k$ is $\widetilde{A} = A \cup (\bigcup_{i=1}^d S_i) \cup \{1, \ldots, d\}$ of size $\max\{|A|, W, d\}$ compared to $|A|W^d n^d$ in our original construction. Moreover, the state space is $\widetilde{S} = S \times \{0, 1, \ldots, d(m + 1) + 1\} \times A \times \{1, \ldots, d\}^m \times S \times \{0, 1\}$, which is similar to Section 5 up to the new factors $\{1, \ldots, d\}^m$ that keep the chosen scope.

As in Section 5, a state $s$ is mapped to $(s, 0, \bot)$ and taking action $a \in A$ transitions to $(s, 1, a, \bot)$ while other actions are not legal. When the counter is between $i(m + 1) - m$ and $i(m + 1) - 1$ the legal actions are $\{1, \ldots, d\}$ and the chosen indices are just stored in the state (denote them by $Z$). Then, the legal actions in state $(s, i(m + 1), a, Z, w_1, \ldots, w_{i-1}, \bot)$ are $S_i$, and picking action $w \in S_i$ transitions to $(s, i(m + 1) + 1, a, Z, w_1, \ldots, w_{i-1}, w_i, \bot)$ with probability

$$\bar{P}_{i,Z}^k(w_i \mid s[Z], a) - \mathcal{W}_{i,Z}^k(w_i \mid s[Z], a) + \mathbb{I}\{w_i = w\} \cdot \sum_{w' \in S_i} \mathcal{W}_{i,Z}^k(w' \mid s[Z], a).$$

At this point the validating bit also checks that $Z$ is consistent for factor $i$, and turns to $0$ if not. Finally, we transition from $(s, d(m + 1) + 1, a, Z', w_1, \ldots, w_d, b)$ deterministically to $(s', 0, \bot)$, where $s' = (w_1, \ldots, w_d) \in S$.

Just like Section 4.2, the transition function of $\widetilde{M}^k$ is no longer factored because some scopes include the entire state-action space. However, as we previously showed, we can overcome this and perform the optimistic transition according to a selected scope while maintaining small scope size by constructing the FMDP $\widehat{M}^k$ with a "temporary" work space $\Omega^m$, where $\Omega = \omega^n \times \omega^{n/2} \times \cdots \times \omega^2 \times \omega$. Notice that it is much smaller now because we are not performing the transition for all $d$ factors simultaneously. Thus, the oracle needs to solve an FMDP with scope size $m + 4$, number of factors $2d + m + 3 + 2nm$, size of each factor bounded by $\max\{W, |A|, d(m + 1) + 2, n\}$ and small number of actions.

Finally, a similar construction to Section 5 can be used in order to bound the regret. It involves the MDP $M'$ with state space $\widetilde{S}$, that stretches each time step of $M$ for $2 + d(m + 1)$ steps but models the exact same process as $M$.

**Theorem 6.** *Running NFA-SLF-UCRL on a factored MDP with non-factored actions and unknown structure ensures, with probability at least $1 - \delta$,*

$$Reg_T(M) = \widetilde{O}\bigg( \sum_{i=1}^d \sum_{Z:|Z|=m} D\sqrt{|S_i||S[Z_i^P \cup Z]||A|T} + \frac{1}{\ell} \sum_{j=1}^\ell \sum_{Z:|Z|=m} \sqrt{|S[Z_j^r \cup Z]||A|T} \bigg).$$

## F   Lower Bound

We associate an independent multi-arm bandit (MAB) problem to every tuple $(i, w_1, \ldots, w_m) \in \{1, \ldots, d\} \times \{1, \ldots, W\}^m$. Without loss of generality we assume that the rewards of all the MABs are either $0$ or $1$.

Now we construct the following factored MDP $M = (S, A, P, R)$, where the state space is $S = \{0, 1, \ldots, \log d + 1\} \times \{0, 1\}^{\log d} \times \{0, 1, \ldots, W\}^d \times \{0, 1\}^d \times \{0, 1\}^{d/2} \times \cdots \times \{0, 1\}^4 \times \{0, 1\}^2$, and the action space is non-factored of size $|A|$. Note that the state space has $3d + \log d$ factors with maximal size $\max\{W + 1, \log d + 2\}$.

The idea is to split the $T$ time steps into blocks of $2 + \log d$ steps. In each block the agent faces a randomly chosen MAB problem (out of the $dW^m$ independent MABs). We make sure that it cannot infer anything about the different MABs, and thus must solve them sequentially. Since the $t$ steps lower bound for each MAB is $\Omega(\sqrt{|A|t})$, and the expected number of times that the agent faces each MAB is $\frac{T}{dW^m(2+\log d)}$, the total regret is

$$\Omega\Big(\sum_{i=1}^{d} \sum_{v \in \{1,\ldots,W\}^m} \sqrt{|A| \frac{T}{dW^m(2 + \log d)}}\Big) = \Omega\Big(\sqrt{\frac{d}{\log d} W^m |A| T}\Big).$$

We do not make the full formal argument about the relation between the lower bound and the expected number of times we encounter each MAB, but it can be found in the lower bound proof of Rosenberg et al. [2020] for example.

We now continue to define the FMDP that makes the agent face the MABs sequentially. There is only one reward factor. Its scope is the last two bits and the first factor (the counter). It gives a reward of $1$ only when the counter is $\log d + 1$ and the last two bits contain a $1$. Otherwise the reward is $0$.

The transition function is defined as follows:

- The first factor is called the counter factor. It counts deterministically modulo $\log d + 2$.

- The next $\log d$ bits are called the location bits, and they determine the location of the MAB within the state. Each bit $j$ of these $\log d$ location bits is simply changing uniformly at random, i.e., becomes $0$ or $1$ with probability $1/2$.

- The next $d$ factors are called the value factors, and they give the MAB instance that is encountered by the agent at this time block. The transitions for the $i$-th value factor are defined as follows. When the counter is $0$ denote by $x \in \{1, \ldots, d\}$ the integer that the $\log d$ location bits represent. If $x \le i < x + m$ this factor is chosen uniformly at random from $\{1, \ldots, W\}$ and otherwise it is $0$. When the counter is larger than $0$ this factor is just $0$. Note that the scope size for these factors is $\log d + 1$.

- The next $d$ bits are called the reward bits, and they represent the rewards given by the MABs. The transitions of the $j$-th reward bit is defined as follows. When the counter is $1$ denote by $(w_1, \ldots, w_m)$ the values of factors $j$ to $j + m - 1$ of the $d$ value factors. If one of them is $0$ than the $j$-th reward bit is zero, and otherwise its value is determined by the reward of MAB $(j, w_1, \ldots, w_m)$. When the counter is not $1$ this factor is just $0$. Note that the scope size for this factor is $m + 1$. Moreover, this is the only MAB instance that the agent gets any information about, which forces it to solve all the MABs sequentially.

- The final bits $\{0, 1\}^{d/2} \times \cdots \times \{0, 1\}^4 \times \{0, 1\}^2$ take the $d$ reward bits and extract whether they contain a $1$ or are all $0$. Notice that this encodes exactly the reward given by the current MAB. Similarly to the SLF-UCRL algorithm, this can be achieved with scope size $3$ (each bit needs to consider two bits from the previous layer and the counter) and within $\log d - 1$ steps. This is done when the counter is $2, \ldots, \log d$ and then the last two bits contain a $1$ if the answer is yes, and are both $0$ if the answer is no.

**Remark** (Dependence in the diameter)**.** Our main goal in the lower bound was to show that polynomial dependence in the number of factors and exponential dependence in the scope size are necessary. This was not clear from previous lower bounds as they used scopes of size $1$, and did not have a dependence on $d$ (because there was an average over factors). Therefore, we did not get a dependence on the diameter $D$. While getting the dependence in $D$ might be tricky in the average-reward setting,

it is straightforward to get a $\sqrt{H}$ dependence in the finite-horizon setting (with horizon $H$). In the finite-horizon setting our construction is similar such that in each episode one MAB is faced and the agent earns the same reward for $H - (\log d + 2)$ steps (after the reward is chosen in the first $\log d + 2$ steps, the agent has no control and just keeps receiving the same reward). This gives a lower bound of $\Omega\left(\sqrt{\frac{d}{\log d} H W^m |A| T}\right)$ that matches the upper bound of Chen et al. [2021] (up to logarithmic factors), thus proving that this is indeed the minimax optimal regret.

# G  Experiments

The code is available here:

https://github.com/avivros007/Factored-MDP-with-Unknown-Structure.

We perform numerical experiments to support our theoretical claims regarding the SLF-UCRL algorithm. The experiments are performed on the *SysAdmin* domain [Guestrin et al., 2003]. This domain consists of $N$ servers that are organized in a graph with a certain topology. Each server is represented by a binary variable that indicates whether or not it is working. At each time step, each server has a chance of failing, which depends on its own status and the status of the servers connected to it. There are $N + 1$ actions: $N$ actions for rebooting a server (after which it works with high probability) and an idle action. In previous work [Guestrin et al., 2003, Xu and Tewari, 2020, Talebi et al., 2021], researchers have performed experiments with two different topologies: A circular topology in which each server is connected to the next server in the circle, and a star topology in which the servers are organized in a tree with three branches.

In each topology, the status of each server depends on at most one other server (and its own status and the action) so the scope size is $m = 3$. The number of state factors is $d = N$, the size of each state factor is $W = 2$, the action space is of size $|A| = N + 1$. Thus the state-action space is of total size $|S \times A| = 2^N (N + 1)$ which is exponential in the number of servers $N$, while the representation of this FMDP is only polynomial in $N$.

In our experiments, we set $\delta = 0.01$ and report for each domain the average results over $10$ independent experiments (and the standard error in the shaded area). Our code is based on the code of Talebi et al. [2021] which was made publicly available via https://github.com/aig-upf/dbn-ucrl. To that code we added a new class called SLFUCRL that implements our algorithm, i.e., maintains sets of consistent scopes (we focus on transitions and assume that the reward scopes are known) and integrates them within the optimistic policy computation. For the planning oracle, we simply solve the full optimistic MDP using extended value iteration (up to some error). We note that for finite-horizon we could solve the optimistic MDP exactly.

Figure 2 shows that in a variety of scenarios the SLF-UCRL algorithm acts as predicted by our theoretical guarantees. In (a),(b),(c) we used the circular topology with $N = 4, 5, 6$ servers, respectively, and in (d) we used the star topology with $N = 4$ servers. We can see that SLF-UCRL eliminates the wrong scopes, and that its regret is comparable to that of the Factored-UCRL algorithm [Osband and Van Roy, 2014] that has full knowledge of the factored structure in advance. Moreover, the regret of SLF-UCRL is significantly better than that of the UCRL algorithm [Jaksch et al., 2010] that simply ignores the existence of a factored structure, demonstrating the importance of learning the structure (as the SLF-UCRL algorithm does). "SLF-UCRL$i$" refers to $i$ factors whose scope needs to be learned, demonstrating that additional domain knowledge can be easily integrated into the SLF-UCRL algorithm and help it both in terms of regret and in terms of computational complexity (which does not appear in the graphs).

Note that for experiment (a) we used a slightly stricter threshold (by a factor of $10$) to eliminate inconsistent scopes, but then we saw that we can eliminate them faster without eliminating the true scopes. This is why it takes $20000$ steps (and not $15000$) to eliminate all scopes in experiment (a).

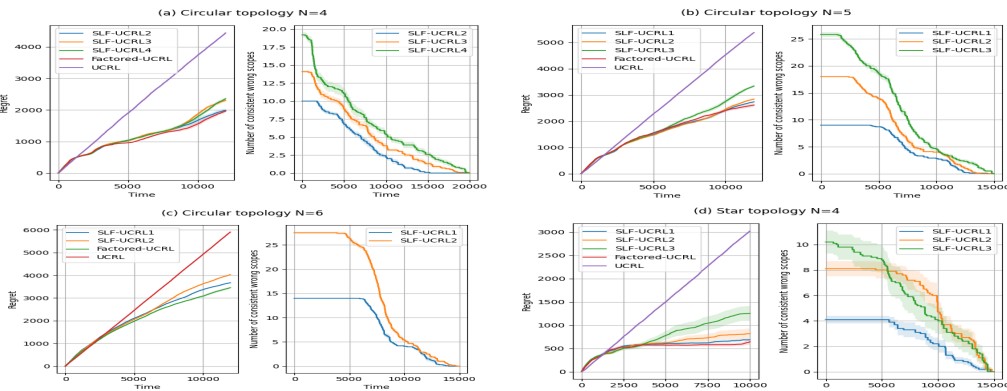

Figure 2: SLF-UCRL performance on *SysAdmin* domain.