# OpenReview forum: "Oracle-Efficient Regret Minimization in Factored MDPs with Unknown Structure"
_NeurIPS.cc/2021/Conference — NeurIPS 2021 Poster_

### Official Review · Reviewer_nfbV · 2021-07-11

**Rating:** 7
**Confidence:** 3

**Summary:**

This work concerns reinforcement learning in factored MDPs. The first part of the work combines previous work on regret minimization in FMDPs using a planning oracle (following prior work by Osband and Van Roy and Xu and Tewari) with work on structure learning in FMDPs. In this context, structure learning means the following: in order for FMDPs to possess compact transition and reward functions, one typically assumes they are represented by graphical models in which each factor depends on a limited number of factors from the previous step; the structure learning problem is essentially to identify these dependencies. Here, Xu and Tewari's algorithm is modified so that it identifies the structure during learning, while maintaining its regret bound. The previous works in this area provided sample complexity bounds rather than regret bounds.
In addition to the regret bounds, experiments are provided, demonstrating that the algorithm is somewhat competitive with the algorithm of Osband and Van Roy that is provided with the dependency structure.

In more detail, as in the algorithm of Xu and Tewari, this algorithm uses the idea of augmenting the actions that originally appeared in Jaksch et al.; essentially, in addition to allowing the planner to select an action in the actual MDP, Xu and Tewari allow the planner to select one of the extremal transition functions. Here, the possible dependency structures are added to the options presented to the planner.

An additional transformation is proposed that converts the resulting large action space to a sequence of moves in an augmented state space. The same transformation could be applied to the algorithm of Xu and Tewari as well as the unknown structure algorithm, so this could be considered a second, separate part of the work.

The final part of the work is an improved lower bound that in particular shows that regret must depend exponentially on the scope sizes, and asymptotically matches the regret obtained by some recent works (e.g., Tian et al. 2020)


**Limitations And Societal Impact:**

This is fine.

**Main Review:**

The lower bound is probably more significant than the upper bound, and I wish more detail on this had been included in the body of the paper. I see that it is using the ideas developed for the upper bounds, but I would have liked the construction to be sketched in some detail.

The ideas underlying the algorithm are similar to those underlying the earlier works that gave sample complexity bounds rather than regret bounds, just invoked in the context of Xu and Tewari's construction. So there actually is not a ton of novelty there, and the algorithm simply represents all of the n choose m possible dependency sets explicitly. The analysis of dependency sets is similar to the reasoning used by Strehl et al. in one of the early works in this direction. The novelty essentially lies in converting it back to a factored MDP, since allowing the planner to choose the scope creates dependencies with all of the possible scopes. This is accomplished by a second transformation involving an augmented state space that collects the selected scope by a series of local transitions.

I do appreciate that the work includes an experimental evaluation of this algorithm, as otherwise I would worry about the effect of the reduction on the planners. Just because the planners are adequate for the handful of benchmarks doesn't imply that they can solve arbitrary FMDPs efficiently. For example, when SAT-solvers are run on the CNFs obtained by reducing from hard problems, suddenly they aren't so "effective in practice." I wish the plot were large enough that it could be read on a printed page, though. Also, I assume that the reason that the comparison is given to Osband and Van Roy is that the factored action set constructed by Xu and Tewari's algorithm can't be handled by the available planners. But then I also wonder how the structure learning algorithm would compare to the result of the second transformation (the algorithm NFA-DORL described in Theorem 2).

In summary, although the "main result" here seems to be largely a straightforward combination of existing techniques, the state-space transformation that underpins it, as well as the reduction to non-factored actions and the lower bound does seem to be interesting and yields some interesting results. I'd honestly have preferred a paper that didn't treat the unknown structure problem in such detail, in favor of a more thorough presentation of the lower bound.



**Time Spent Reviewing:**

3

---

> ### Author Response · Authors · 2021-08-08
> **Response to Reviewer nfbV**
>
> We thank the reviewer for the insightful review. While we agree that our lower bound is an important result (and we will add more details about it in the main text), we argue that the unknown structure setting is also challenging and important for many applications and that we present in this paper many novel techniques that are not simple combinations of previous work.
>
> First, combining consistent scopes as actions for the optimistic MDP appears in this paper for the first time. Moreover, a naive combination results in an optimistic factored MDP with huge scope size (Section 4.1), and the construction that converts this optimistic MDP into a factored MDP with small scope size (Section 4.2) is novel and important in order to ensure that the algorithm is oracle-efficient. Otherwise, the oracle would need to solve a factored MDP with scope size that is equal to the number of factors (i.e., a tabular MDP) which is obviously going to be exponential in the number of factors. Thus, the novel transformation described in Section 4.2 is crucial!
>
> Second, the model of FMDPs with non-factored actions is considered in this paper for the first time, and is important to mitigate between the planning literature and the regret minimization literature. Extending the SLF-UCRL algorithm to this model requires additional novel techniques (like stretched time steps), that also prove to be useful in the construction of the lower bound.
>
> We take your advice and will add more details about the lower bound construction and its proof technique in the main text. Please keep in mind that the techniques from our algorithm for unknown structure are proved to be useful for the construction of our new lower bound. This is another reason why our contributions to the unknown structure setting are important (which is obviously secondary to the fact that we present the first regret bound for this important setting).
>
> We note that in the experiments we used the naive oracle that simply solves the optimistic MDP as if it was not factored. For this reason, the algorithm of Osband and Van Roy is similar to the algorithm of Xu and Tewari. We chose to compare our algorithm to that of Osband and Van Roy because it is a little easier to implement. Since we prove in the paper that the transformation does not change the optimal policy, it does not matter if we perform the transformation or simply solve the optimistic MDP as is (of course it matters computationally).

---

### Official Review · Reviewer_shHj · 2021-07-14

**Rating:** 6
**Confidence:** 3

**Summary:**

This paper studies regret minimization in factored MDPs with unknown structure. The algorithm combines optimism in the face of uncertainty principle and a structure learning approach. With planning oracles, both the sample and computation complexity depend polynomially on the problem size. This paper also provides a regret lower bound and empirical evaluations.



**Limitations And Societal Impact:**

The authors adequately addressed limitations and potential negative societal impact.

**Main Review:**

This paper is well written and the authors clearly discussed related works. The SLF-UCRL algorithm is the first algorithm that achieves \sqrt{T} regret in factored MDPs with unknown structure. The concept of consistent scopes is novel and intuitive. Ignoring the computation complexity, introducing consistent scopes already gives a \sqrt{T} regret algorithm, whose analysis is simple and easy-to-follow. Overall, I believe the theoretical results are sound.

The SLF-UCRL algorithm addresses the difficulty of structure learning – deciding the scope of each variable – by enumerating all possible scopes and checking consistency. The key observation is that the number of possible scopes is bounded by n^m, which is in the same order as the encoding size (W^m). The n^m dependence appears not only in the computation complexity but also in the regret. On the other hand, the regret lower bound is W^m (although the lower bound doesn’t characterize the hardness of unknown structure). When n>>W the regret bound could be suboptimal. It is unclear whether this dependence is an artifact of the algorithm (or the analysis), or unavoidable for Alg. 2.

Regarding the computation complexity, the algorithm heavily relies on the computation oracle. In fact, the oracle needs to “pick the scopes as well as the actions”. Since computing the optimal policy of a given factored MDP is already hard, it is unclear to me whether the reduction increases the hardness of planning furthermore. During the reduction, some of the nice properties that may guarantee efficient planning could be lost. In other words, it seems to me that the authors essentially sweep all the computation tasks into the planning oracle.


**Time Spent Reviewing:**

4

---

> ### Author Response · Authors · 2021-08-08
> **Response to Reviewer shHj**
>
> We thank the reviewer for the thorough review.
>
> We would like to note that $n^m$ computational dependence is unavoidable since this is the number of possible scopes (without any additional knowledge). This dependence also seems unavoidable in terms of regret, but we agree that the dependence may be additive and not multiplicative, i.e., $\sqrt{W^m T} + n^m$ and not $n^m \sqrt{W^m T}$. This is left for future work.
>
> Regarding the computation oracle, it is important to note that the nice properties that the MDP may have cannot be exploited without knowledge of the structure! Moreover, even with known structure, the method of Xu and Tewari [2020] introduces changes to the MDP so nice properties will be lost there too (and of course other methods cannot even utilize the oracle, and must use exponential computation). Finally, once a lot of scopes are eliminated (due to inconsistency), the optimistic MDP becomes closer and closer to the real one (for example, in the experiments this happens quickly).

---

### Official Review · Reviewer_v8hf · 2021-07-16

**Rating:** 7
**Confidence:** 3

**Summary:**


This work studies the regret minimization problem of non-episodic factored stochastic MDPs with unknown structure, where the state and action space can be factored as products of several sets. With assumptions on the transition and reward function based on the unknown structure of state-action space (unknown factorization), the authors propose the first oracle efficient algorithm which achieves sqrt(T) regret, and prove a lower bound of learning factored MDPs with known transition.

**Limitations And Societal Impact:**

This work is pure theoretical and does not have any potential negative societal impact.

**Main Review:**


1. Contribution (Strengths)

The main technical contribution is the novel reduction method described in Section 4.1 and 4.2. First, the authors follow the idea of the framework of optimism in face of uncertainty, that is, the learner computes the confidence width which constructed based on the concentration inequalities on the transition and reward function. Then, the learner constructs an optimistic factored MDP with extended state action space, and embeds the optimistic selection of transition functions into the action space, that is, the action now decides both the transition and the true action. Note that, since the constructed instance is factored MDP with bounded scope size, the policy can be computed by the oracle solver.

The authors also propose the first lower bound for regret minimization of factored MDPs, following similar ideas of Rosenberg et al (2020) which constructs a lower bound for stochastic shortest path.

Though I don't acknowledge that the pure theoretical results should be validated by experiments, the authors perform experiments on SysAdmin domain and show that the proposed algorithm is practical, in the sense that it achieves comparable performance with the state of the art (that is, Factored-UCRL with known structure).

2. Weaknesses

Other than the reduction method, the other designs such as sequentially removing inconsistent scopes (strucuture, factorization) are standard and expected.

3. Writing Issues

423, robabilistic->probabilistic

693, regert->regret

**Time Spent Reviewing:**

48

---

> ### Author Response · Authors · 2021-08-08
> **Response to Reviewer v8hf**
>
> We thank the reviewer for the thorough review. We want to emphasize that many of the techniques proposed in the paper are not standard and involve novel ideas.
>
> First, combining consistent scopes as actions for the optimistic MDP appears in this paper for the first time. Moreover, a naive combination results in an optimistic factored MDP with huge scope size (Section 4.1), and the construction that converts this optimistic MDP into a factored MDP with small scope size (Section 4.2) is novel and important in order to ensure that the algorithm is oracle-efficient. Otherwise, the oracle would need to solve a factored MDP with scope size that is equal to the number of factors (i.e., a tabular MDP) which is obviously going to be exponential in the number of factors. Thus, the novel transformation described in Section 4.2 is crucial!
>
> Second, the model of FMDPs with non-factored actions is considered in this paper for the first time, and is important to mitigate between the planning literature and the regret minimization literature. Extending the SLF-UCRL algorithm to this model requires additional novel techniques (like stretched time steps), that also prove to be useful in the construction of the lower bound.
>
> Finally, our lower bound features a completely new construction that uniquely exploits the factored structure of the FMDP. This lower bound improves upon previous constructions that are much more standard, and tightly matches the best known upper bound for episodic FMDPs. The only thing that relies on the lower bound of Rosenberg et al. [2020] is the proof idea (which builds on the lower bound for multi-arm bandit similarly to other lower bounds in RL).

---

### Official Review · Reviewer_xFpj · 2021-07-18

**Rating:** 7
**Confidence:** 4

**Summary:**

This paper studies regret minimization in Factored MDPs (FMDPs) under the average-reward criterion, and without assuming prior knowledge on the factorization structure. The authors present the SLF-UCRL algorithm, a UCRL2-style algorithm that borrows some elements from DORL, and analyze its regret. The authors also study the case of FMDPs with non-factored action-space and present an algorithm for this setup. This is important to ensure compatibility with most known approximate efficient planners for FMDPs. Yet another worth noting contribution of the paper is to present a regret lower bound for the case of known structure, which improves upon existing ones in the literature and matches recent regret upper bounds in the episodic setting.


**Limitations And Societal Impact:**

See above.

**Main Review:**

Learning in FMDPs constitutes an interesting and important probelm in RL. FMDP models were introduced more than two decades ago and some relevant RL algorithms have been around since the early days of FMDPs. Nonetheless, regret minimization in FMDPs dates back to only a few years ago. It has recently re-gained attention by the research community, in both episodic and average-rewad settings. To my knowledge, this work is the first studying regret minimization in FMDPs without the knowledge of factorization structure. Although the key algorithmic idea for learning the structure is from existing literature, I believe combining that with the optimistic principle and deriving a rate-optimal regret is non-trivial and challenging. The derived regret bound is significant as it avoids an exponential dependence on $d$ (albeit exponentially dependent on $m$).

The paper is written very well, typo-free, and is well-structured. There are however a few minor comments and suggestions to further improve the writing (see below). The paper is also written mostly clearly and precisely. I was unable to check the proofs in details in this limited review period, but they appear correct to me. Below, I provide some detalied comments, which seem crucial to be addressed.

Detailed Comments:

1- The presented lower bound is interesting and informative, and in my opinion, it improves our understanding about the true difficulty of FMDP learning. A missing piece here is the absence of $D$-dependencies, which is not discussed much in the main text. I may ask the authors to remark this clearly and precisely in the main text, in both the introduction and Section 6, though lengthy discussions could be provided in the appendix.

2- Although the paper is mostly clearly written, I found part of the description of SLF-UCRL in Section 4.2 a bit confusing. I understand the procedure is difficult to explain, but tend to think that there might be alternative ways to present that would be easier to follow. Also, providing some illustration or exemplifying on a toy FMDP could prove very helpful.

3- I may also ask the authors to further elaborate on the notion of “stretched steps” introduced in Section 4.2.

4- Simulation results: The figures are not legible unfortunately. This also applies to those reported in Appendix G, although there is no space limit for the supplementary. I may ask the authors to also consider reporting the corresponding confidence intervals.

5- Simulation results: In all experiments, UCRL2 suffers from a linear regret (for the chosen horizons $T$). If this is due to rather short time horizon, why not running the experiments for large enough horizons (at least for some experiment if it is time consuming)? Otherwise, the reason could be emphasized.

Overall I found the presented algorithms and results of enough significance, and assuming that the authors will address the above comments, vote for a score of 7.

Minor Comments and Suggestions:

Line 44 (and elsewhere): factored MDPs => FMDPs  --- The acronym FMDP is already introduced in page 1, and could be used in the rest.

Line 87: the expected gain => the gain --- By definition (e.g., in line 79), the gain considers expected cumulative reward (as opposed to, e.g., the high probability notion, which is rather uncommon).

Line 119: Note that $W^d$ is not the actual size of state-action space, but rather an upper bound on that, which could be tight in some cases.

Line 150: UCRL => UCRL2 --- The algorithm in Jaksch et al. (2010) is UCRL2 and not UCRL. More precisely, UCRL is a different algorithm, an earlier version of UCRL2, presented in (Ortner & Auer, 2007).

Line 245: … is replaced with => is replaced by (because of passive form. Otherwise “with” is correct)

Line 238: Similarly to the transitions, … are … => Similar to … (because of “are”)

**Time Spent Reviewing:**

5 hours

---

> ### Author Response · Authors · 2021-08-08
> **Response to Reviewer xFpj**
>
> We thank the reviewer for the kind review and useful comments. All the minor comments will be fixed.
>
> 1. This is a good comment. Indeed the lower bound dependence in the diameter is not clear enough in the main text (but is explained in the appendix). Our lower bound gives the correct dependence in the horizon $H$ in the case of finite-horizon FMDPs, but does not give dependence in $D$ in the case of average-reward. As discussed in Xu and Tewari [2020], it is not clear that the regret must depend on $D$ and not on a tighter connectivity measure in the case of factored MDPs. We leave this to future work and close the gap only in the finite-horizon case in this setting. Note that for average-reward the best upper bound is still far from the finite-horizon case anyway. This will be explained in the main text more clearly.
>
> 2. We agree that an illustration could be helpful for explaining the procedure and it will be added in the camera-ready version.
>
> 3. Stretched time steps allow us to extract the relevant scopes from the current state in order to perform the transition while maintaining small scope sizes in the optimistic MDP. This is crucial in order to keep our algorithm oracle-efficient (since this keeps small scope size), and also plays an important role in the construction of our lower bound. Since this is a key technique in the paper, to make sure that it is clear, we will add an intuitive explanation of the technique in the beginning of Section 4.2 before diving into the details.
>
> 4. Confidence intervals will be added to the figures, and they will be enlarged.
>
> 5. The regret of UCRL2 grows linearly with the number of states, which is exponentially larger than the regret of algorithms that take into account the factored structure. Thus, in order to observe sub-linear rate the number of steps must be at least as large as the number of states. This is not realistic and this is exactly what the experiments show. If this is helpful, we can take the smaller experiments and run them for a much larger number of time steps.

---

> > ### Comment · Reviewer_xFpj · 2021-08-27
> > **Response to Rebuttal**
> >
> > I have read the other reviews and the rebuttal, and would like to thank the authors for their responses to my comments.
> >
> > Incorporating the suggested changes into the revised version sounds an excellent idea. You may skip the comment no. 5, keeping UCRL2's regret as is, but adding a brief explanation similar to the one in the rebuttal could be helpful.
> >
> > Overall, this is a solid and well-executed paper addressing a challenging and interesting problem, and I therefore maintain my current score (of 7) and confidence (of 4).

---

### Decision · Program_Chairs · 2021-09-27

**Decision:**

Accept (Poster)

**Comment:**

The paper presents the first oracle-efficient algorithm for reinforcement learning factored MDPs when the factored structure is unknown. This is a nice addition to the literature on reinforcement learning with large state spaces, and all reviewers were positive on the paper, so I recommend acceptance. Please incorporate any comments from the discussion period into the final manuscript.